neuroscience, behaviour, physiology

mechanoreceptor channel, SA mechanoreceptors, RA mechanoreceptors, Szechuan pepper, hydroxy-α-sanshool, tactile perception

**Author for correspondence:**
Patrick Haggard
e-mail: p.haggard@ucl.ac.uk

†Equal contribution.

# Touch inhibits touch: sanshool-induced paradoxical tingling reveals perceptual interaction between somatosensory submodalities

Antonio Cataldo[1,2,3,†], Nobuhiro Hagura[4,5,†], Yousef Hyder[1,4] and Patrick Haggard[1,2]

[1]Institute of Cognitive Neuroscience, University College London, Alexandra House 17 Queen Square, London WC1N 3AZ, UK
[2]Institute of Philosophy, School of Advanced Study – University of London, Senate House, Malet Street, London WC1E 7HU, UK
[3]Cognition, Values and Behaviour, Ludwig Maximilian University, Gabelsbergerstraße 62, 80333 München, Germany
[4]Center for Information and Neural Networks, National Institute of Information and Communications Technology, 1-4 Yamadaoka, Suita City, Osaka 565-0871, Japan
[5]Graduate School of Frontier Biosciences, Osaka University, Osaka, Japan

AC, 0000-0003-1228-8577; PH, 0000-0001-7798-793X

Human perception of touch is mediated by inputs from multiple channels. Classical theories postulate independent contributions of each channel to each tactile feature, with little or no interaction between channels. In contrast to this view, we show that inputs from two sub-modalities of mechanical input channels interact to determine tactile perception. The flutter-range vibration channel was activated anomalously using *hydroxy-α-sanshool*, a bioactive compound of Szechuan pepper, which chemically induces vibration-like tingling sensations. We tested whether this tingling sensation on the lips was modulated by sustained mechanical pressure. Across four experiments, we show that sustained touch inhibits sanshool tingling sensations in a location-specific, pressure-level and time-dependent manner. Additional experiments ruled out the mediation of this interaction by nociceptive or affective (C-tactile) channels. These results reveal novel inhibitory influence from steady pressure onto flutter-range tactile perceptual channels, consistent with early-stage interactions between mechanoreceptor inputs within the somatosensory pathway.

## 1. Introduction

The sense of touch involves neural processing of multiple features of cutaneous stimuli. Features extracted from stimuli to the skin are conveyed to the brain through distinct classes of afferent fibre [1,2]. Some fibres are tuned for specific spatio-temporal skin deformation patterns, and are considered mechanoreceptor channels, while others are tuned for thermal and noxious features [3,4]. These neurophysiological channels can also be studied psychophysically, because different qualities of sensation (flutter, high-frequency vibration, steady pressure etc.) are thought to be conveyed by each afferent class [1,2,5].

Although the characteristics of each perceptual channel have been explored, little is known about how the information from each channel interacts to provide an overall sense of touch. For example, inhibitory interaction between mechanical and pain/thermal channels has been well established [6,7], but it is still unclear whether similar inhibitory interactions

occur between the different mechanoreceptor channels or 'submodalities'. Classical accounts assume that each mechanoreceptor channel (RA, SA1, PC, SA2) carries *independent* information about specific tactile features [8,9], and that this independence is preserved in early cortical somatosensory processing [10–13]. The independence hypothesis has been recently challenged by neurophysiological studies of responses in single neurons. These studies suggested interaction between signals from different mechanoreceptor channels at spinal, thalamic and cortical levels [2,14,15]. However, to our knowledge, few psychophysical studies have investigated the implications of inter-channel interaction for perception, as opposed to neural coding.

Here, we show, to our knowledge, the first *human psychophysical* evidence that signals from different mechanical feature channels do indeed interact to determine tactile perception. Specifically, we show that perception of flutter-range mechanical vibration (mediated by a perceptual channel putatively corresponding to a rapidly adapting (RA) neurophysiological channel) is inhibited by concurrent activation of the perceptual channel for steady pressure (putatively corresponding to a slowly adapting (SA) channel). Thus, 'touch inhibits touch', in a manner similar to the established inhibitory interaction between mechanoreceptive and nociceptive channels (i.e. 'touch inhibits pain') [6,7].

Testing for interaction between perceptual channels might logically involve psychophysical tests of frequency-specific stimuli both alone, and in combination. However, delivering pure frequency-resolved stimuli to mechanoreceptors is difficult, because of the complex propagation of mechanical stimuli through the skin [16]. Here, we take an alternative approach that avoids the difficulties of delivering multi-channel mechanical stimuli, by *chemically* activating one target tactile feature channel, and then measuring the resulting percept in the presence or absence of additional mechanical stimulation to a second channel. In particular, we activated the perceptual flutter-range vibration channel (corresponding to a putative RA channel) using *hydroxyl-a-sanshool*, a bioactive compound of Szechuan pepper (hereafter sanshool) that produces localized tingling sensations with distinctive tactile qualities.

Others have previously demonstrated that sanshool activates the light touch RA fibres [17–20], and we have previously shown that indeed, the perceptual flutter-range tactile feature channel is activated by sanshool [21,22]. Here, we report the perceptual effects of first inducing sanshool-tingling, and then additionally applying controlled sustained pressure (corresponding to the putative SA channel input) to the same skin region. We used psychophysical methods to investigate how the intensity of sanshool-induced tingling sensations was modulated by the additional steady pressure input. This allowed us to assess the interaction between the two perceptual channels that are responsible for tactile steady pressure and tactile flutter features. Two control experiments (electronic supplementary material) reinforced our interpretation that attenuation of tingle by pressure reflects a 'touch inhibits touch' interaction, by ruling out the possibility that sanshool-induced tingle sensation is mediated by other fibre classes, such as nociceptive C-fibres (electronic supplementary material, experiment 5) or C-tactile fibres (electronic supplementary material, experiment 6).

# 2. Material and methods

## (a) Participants

A total of 42 right-handed participants (age range: 18–38 years) volunteered in experiments 1–4 (experiment 1: 10, two females); experiment 2: 10, five females; experiment 3: 8, six females; experiment 4: 14, 12 females). All participants were naive regarding the experimental purpose and gave informed written consent. All methods and procedures were approved by University College London Research Ethics Committee. See the electronic supplementary material for the inclusion criteria of each experiment.

## (b) Experiment 1

Experiment 1 tested whether the tingling sensation induced by sanshool (putative RA channel activation [17–23]) is modulated by application of sustained light pressure (putative SA channel activation [1,2,24]).

Tingling sensation was induced on the upper and lower lip vermilions by applying sanshool (ZANTHALENE, 20% solution, Indena SPA., Milan, Italy) using a cotton swab (figure 1*a*). This stimulation site was chosen because of its dense innervation of mechanoreceptors [25] and thin epidermis [26], which allows the chemical to reach the receptor effectively [21]. Participants sat on a chair, maintaining the upper and lower lip apart by biting a small section of straw between their canine teeth. Each trial started with a baseline period in which participants experienced the tingling sensation and were asked to memorise this baseline intensity. Then, one of eight locations on the lips (figure 1*a*) was manually stimulated by the experimenter with a calibrated probe (diameter: 14 mm, contact force: approx. 1.5 N) for 10 s. The locations touched included three positions each on the upper and lower lip vermillion and two positions above and below the vermillion border, respectively (figure 1*a*). Participants were instructed to always attend to the medial part of the lower lip (position 6 in figure 1*a*), and to judge the intensity of tingling in this specific target location, while the sustained pressure probe contacted one of the eight locations on the lips. Participants rated the tingling sensation relative to the previous baseline. A rating of 10 indicated that the perceived tingling was at the same level as the intensity at baseline; a rating of 0 meant that the participant did not perceive any tingling sensation at all; ratings above 10 would indicate a higher tingling intensity than the baseline period. The rating was given while the mechanical probe remained in stable contact, and 10 s after it had been first applied, to minimize any transient effects of touch onset. An inter-trial interval of a few seconds without mechanical stimulation was always included, to allow the tingling sensation to return. The next trial started only when this was confirmed by the participant. The experiment consisted of six blocks. Each block consisted of 10 trials; two trials each on positions 3 and 6, and one for the remaining six positions (positions 1, 2, 4, 5, 7, 8). This was done to increase sensitivity for the conditions we thought more relevant to the interaction hypothesis. The order of locations for mechanical stimulation was randomised within each participant. The data table for experiments 1–4 can be found in the electronic supplementary material, tables S1–S4.

## (c) Experiment 2

Experiment 2 aimed to replicate and generalize the results of experiment 1. The procedure was largely similar to experiment 1. To make sure that the effect obtained in experiment 1 was not owing to sustained spatial attention to a single target location, participants experienced sanshool tingling all over the lips, and sustained pressure was applied to one of four quadrants (figure 1*b*)

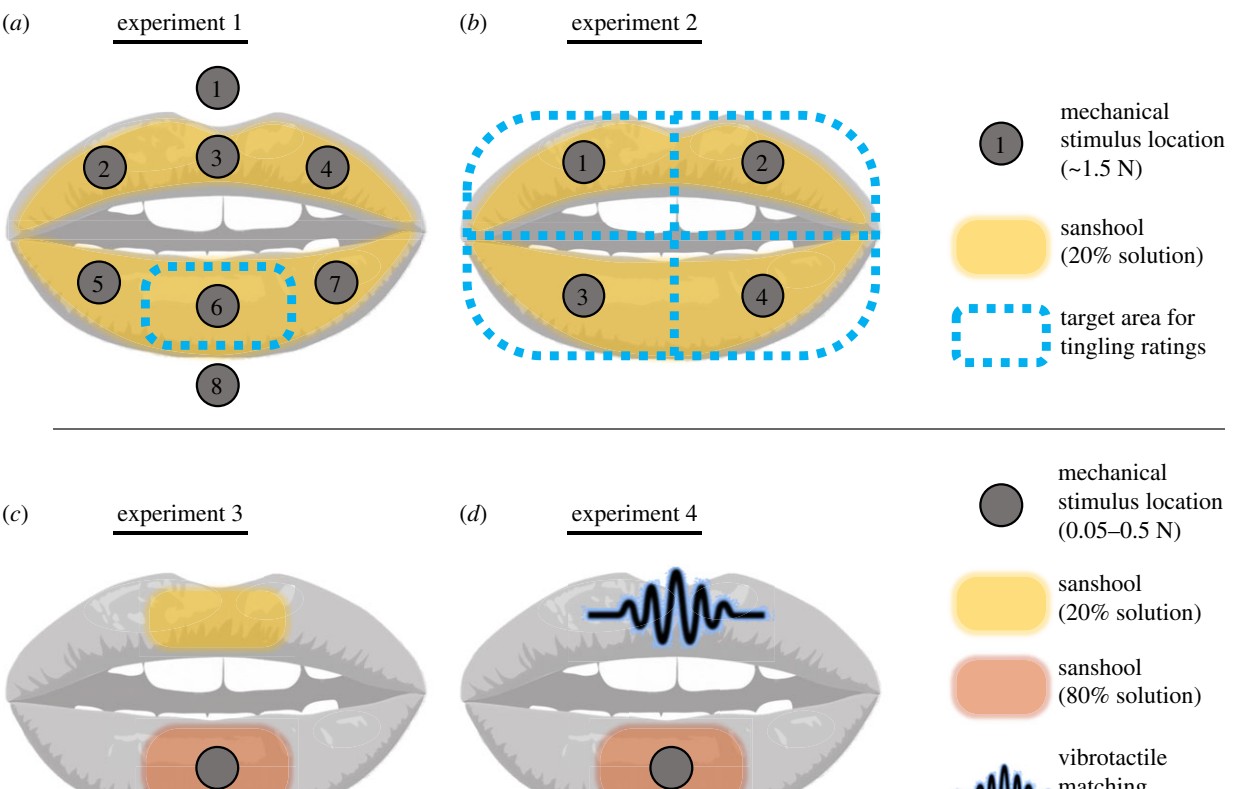

**Figure 1.** Experimental methods. (*a,b*) In experiments 1 and 2, participants experienced sanshool tingling all over the lips, while sustained touch (approx. 1.5 N) was manually applied in different locations for 10 s. Participants reported the effect of touch location on sanshool tingling by rating the change in tingling intensity on the centre of the lower lip (*a*: experiment 1) or all over the lips (*b*: experiment 2). (*c*) In experiment 3, weaker and stronger sanshool solutions caused weaker and stronger tingling intensities on the upper and lower lips, respectively. Different levels of sustained force (0.05, 0.1625, 0.275, 0.375 and 0.5 N) were then applied to the lower lip by a closed-loop robotic device (electronic supplementary material, figure S2). Participants reported which lip had the strongest tingling, as a function of sustained force. (*d*) In experiment 4, participants estimated the intensity of sanshool tingling on the lower lip by adjusting the amplitude of 50 Hz vibration applied to the upper lip until the intensities felt equal. Meanwhile, different levels of sustained force (0.05, 0.20 or 0.35 N) were applied on the lower lip. The adjustment was done at four different timings from the onset of the force (before pressure, and at 0, 5 and 10 s after pressure) (electronic supplementary material, video). (Online version in colour.)

randomly chosen on each trial. This time, instead of only rating the sanshool tingling at a single, fixed location, participants gave separate ratings of tingling intensity for all four lip quadrants, with the order of prompting being randomised. Participants completed six blocks. In each block, sustained touch was applied once to each location (16 ratings).

### (d) Experiment 3

Experiment 3 investigated whether sanshool tingling is modulated by different contact force levels. Given that SA receptor firing is proportional to contact force, [1,2,24], any neural interaction between the putative SA channel and sanshool-evoked tingling (putative RA channel) sensation should produce attenuation of sanshool tingling proportional to contact force. We tested this hypothesis with a novel psychophysical method involving comparing the intensity of two tingle sensations.

First, we arranged a situation where tingling intensity was higher for the lower lip than the upper lip, by applying 80% and 20% concentration sanshool solutions to the lower and upper lip, respectively (figure 1*c*). Participants rested on a chinrest with their lips kept apart. Prior to the main experiment, we confirmed that the stronger solution level of sanshool (lower lip) induced stronger intensity of tingling sensation compared to the weak solution (upper lip) (electronic supplementary material, figure S1). Next, the medial part of the lower lip, which experienced the stronger tingling sensation, was stimulated with different contact forces (0.05, 0.1625, 0.275, 0.375 and

0.5 N). Forces were applied by a closed-loop system comprising a linear actuator (ZABER, XYZ Series, Vancouver, Canada) and a force gauge (Mecmesin, PFI-200N GEB, Slinfold, UK) (electronic supplementary material, figure S2), which continuously maintained the desired pressure level. A cotton bud (diameter 4.5 mm) was placed between the force sensor and the lip. Participants performed a two-alternative forced choice comparison task to indicate whether the upper or the lower lip experienced the more intense tingling sensation. In each trial, one of five different levels of force were applied to the lower lip. One second after the onset of steady pressure, an auditory tone signalled that participants should judge whether the lower or the upper lip currently had the higher intensity of tingling sensation. Participants performed three blocks, each consisting of 10 repetitions of the five contact forces, in random order, giving 150 trials in total.

### (e) Experiment 4

Experiment 4 tested how tingle intensity varied according to the time course of a sustained pressure stimulus. The discharge rate of SA neurons in response to static touch decreases gradually over time, dropping to 30% of the initial firing after 10 s [24]. Therefore, if the activation of the pressure (SA) channel drives suppression of the tingling sensation, sometime-dependent recovery of tingling sensation should occur.

The set-up was similar to experiment 3 (figure 1*d*). However, sanshool (80% solution) was applied on the lower lip only, while the upper lip rested on the probe of a vibro-tactile shaker

(BRÜEL & KJÆR, LDS V101, Nærum, Denmark) (electronic supplementary material, figure S2). In each trial, participants first estimated the intensity of sanshool tingling on the lower lip by adjusting the amplitude of 50 Hz vibration [21] applied to the upper lip until the two intensities felt similar. Amplitude adjustments were made by the participant using the volume setting of an electronic amplifier. The point of perceptual equivalence between mechanical vibration and sanshool-evoked tingle was indicated by pressing a key. Next, one of three different force levels (0.05, 0.20 or 0.35 N) was applied to the lower lip (figure 1d; electronic supplementary material, figure S3, and video S1). An auditory signal was delivered when the closed-loop system had achieved a steady force at the target level. Participants were instructed to note the intensity of the tingling sensation on the lower lip at the time of the beep, and to adjust the amplitude of mechanical vibration to the upper lip until it had a perceptually equivalent intensity. They were instructed to make the adjustment as accurately as possible, while taking no longer than 5 s (mean reaction times: 2.30 s, s.d. 0.62 s). Two further beeps sounded 5 and 10 s after the initial application of sustained force contact, requiring two further matching attempts. Thus, four successive estimations were collected in each trial, one before and three after the pressure application. The experiment consisted of three blocks, with each block consisting of 10 repetitions of the three force levels (90 trials in total). The order of the forces was randomized within each participant.

## 3. Results

### (a) Experiment 1: sustained light-touch (putative slowly adapting input) inhibits sanshool-induced tingling (putative rapidly adapting input)

When the probe was applied at the judged target position (always the centre of the lower lip), tingling intensity was dramatically reduced (to a mean 24.7% ± s.d. 34.0 of the perceived intensity at baseline before the probe was applied) (figure 2a). A one-sample t-test was used to compare the perceived intensity of tingle when the probe was present, to the null mean value of 10 which was defined in our rating scale as the perceived intensity at baseline). The result showed a significant reduction ($t_9 = 7.00$, $p < 0.001$, dz = 2.21; all p-values are Bonferroni-corrected for eight positions).

The tingling sensation at the target position was not affected by pressure on the upper lip or off the lips (all $p > 0.25$, Bonferroni corrected). However, a significant reduction in tingling intensity relative to baseline was found when pressure was applied to the two lower lip locations adjacent to the judged target location (left side: $t_9 = 4.28$, $p = 0.016$ Bonferroni corrected, dz = 1.35; right side: $t_9 = 4.25$, $p = 0.017$ Bonferroni corrected, dz = 1.34). A repeated measures ANOVA showed a clear spatial gradient on the lower, but not the upper lip (electronic supplementary material, Results).

Thus, sustained touch produced a robust inhibition of tingling sensation at the location where the tingling intensity was judged and at adjacent locations.

### (b) Experiment 2: inhibition of sanshool tingling sensation is spatially graded

For the quadrant where sustained touch was applied, we replicated the results of experiment 1, finding robust reduction of tingling during pressure, relative to the baseline (mean rating; 28.3% ± s.d. 36.8 of the baseline intensity) (electronic supplementary material, figure S4). We re-aligned the rating data of each remaining quadrant relative to the quadrant where the sustained touch was applied (figure 2b). We could thus compare the effect on tingling of delivering sustained touch to either the same lip as the location where the tingling rating was judged, or the other lip, and likewise for sustained touch on the same side of the midline as the rated location, or the opposite side. The realigned data showed significant reduction of the tingling rating from the pre-defined baseline value of 10 at the quadrant where the sustained touch was applied ($t_9 = 6.17$, $p < 0.001$ Bonferroni corrected for four comparisons, dz = 1.95), and also when touch was applied at the other quadrant on the same lip ($t_9 = 2.56$, $p = 0.045$ corrected, dz = 1.00), but not when touching the other lip (both $p > 0.26$, uncorrected) (figure 2b).

Next, we directly compared the tingle ratings across different locations in respect to the probe (realigned data). A 2 (lip; same or different to the probe) × 2 (side; same or different to the probe) repeated measures ANOVA revealed significant main effect both for the factor of the lip ($p = 0.003$, $\eta_p^2 = 0.655$) and the side of the probe ($p < 0.001$, $\eta_p^2 = 0.842$), and also an interaction effect ($p = 0.001$, $\eta_p^2 = 0.692$). In the planned comparisons, for the touched lip, the tingle ratings for the touched quadrant was significantly more inhibited compared to the untouched quadrant ($t_9 = 6.30$, $p < 0.001$, dz = 1.99). Interestingly, on the untouched lip also, the quadrant on the same side as the touch again had lower ratings than the other side ($t_9 = 1.58$, $p = 0.025$, dz = 0.50). This implies that the inhibition of the tingling depends on the spatial distance between the location where tingling is judged and the location of sustained touch, both within and across lips. Because the lips did not touch during the experiment (see Methods) this rules out mechanical propagation of sustained pressure as the cause of altered RA mechanoreceptor transduction. Instead, the interaction appears to occur at some neural processing level where afferents from the two mechanoreceptors are integrated in a spatially organised manner.

### (c) Experiment 3: sanshool tingling is parametrically inhibited as a function of contact force

We first checked that sanshool concentration influenced tingling intensity. As expected, participants reported significantly higher intensity for the 80% concentration on the lower lip (average rating: 6.6 ± s.d. 1.55) compared to the 20% concentration on the upper lip (average rating: 3.2 ± s.d. 1.06) ($t_7 = 6.94$, $p < 0.001$, dz = 2.45) (electronic supplementary material, figure S1).

The probability of participants reporting a stronger sensation on the lower lip reduced progressively, and approximately linearly, as the force on the lower lip increased ($F_{1.6,11.2} = 12.09$; $p = 0.002$; $\eta_p^2 = 0.63$) (figure 2d). The suppressive effect of pressure on tingling intensity was confirmed by a significant linear trend analysis ($F_{1,7} = 15.43$; $p = 0.006$; $\eta_p^2 = 0.69$). Thus, RA activation induced by sanshool is parametrically modulated by the signal strength of the SA input.

### (d) Experiment 4: quantifying the relation between sustained force and sanshool–tingling sensation across time

After initial inspection of the data, we found that the distribution of the vibration amplitude matches deviated

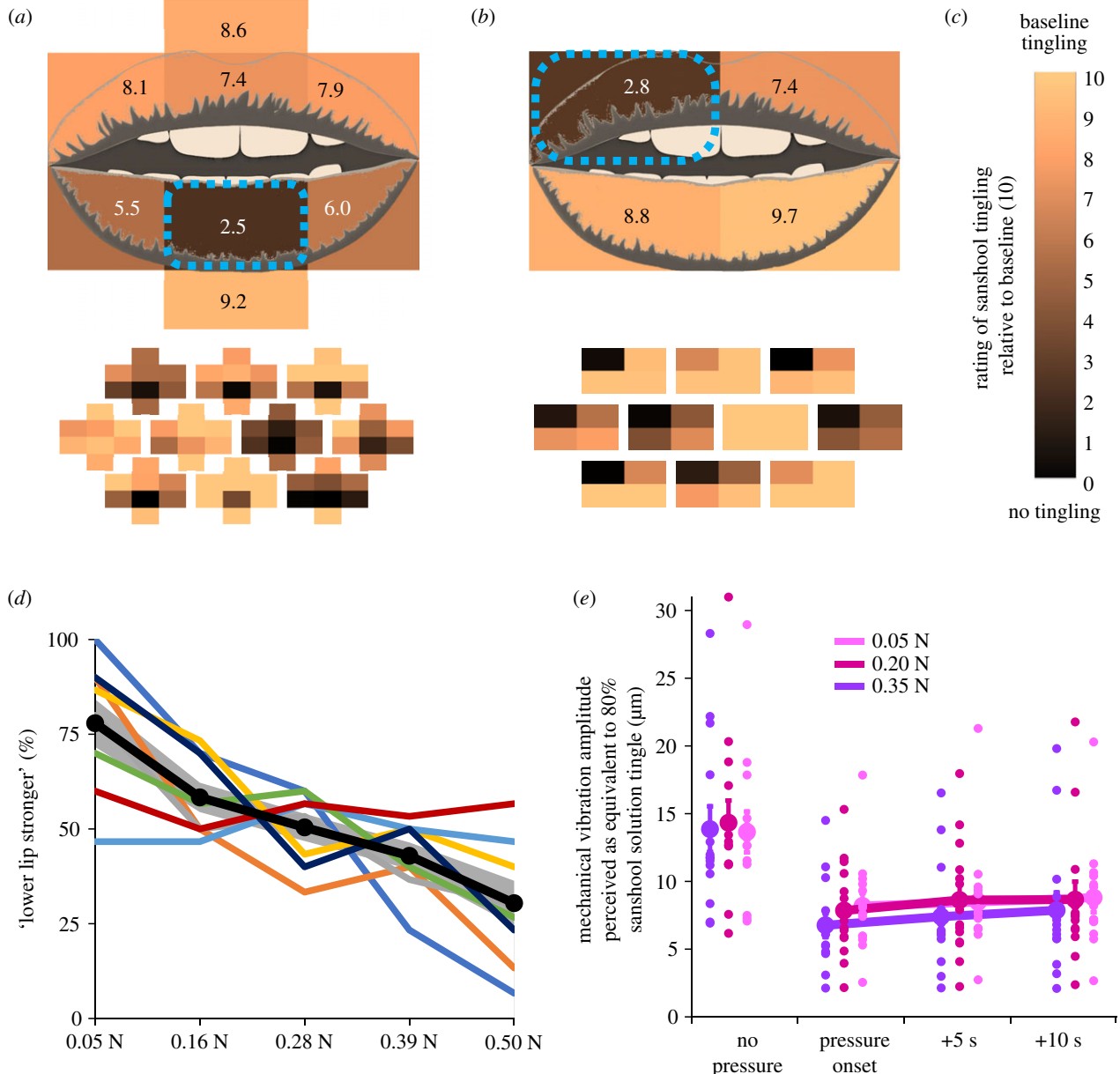

**Figure 2.** Results. (*a–c*) Mean (top) and individual (bottom; *n* = 10 in each experiment) perceived intensity of sanshool tingling as a function of touch location in experiments 1 (*a*) and 2 (*b*). Colour indicates the perceived intensity relative to the baseline period (*c*) (darker colours indicate lower ratings). In experiment 1 (*a*), the perceived intensity of sanshool tingling at the target location (centre of lower lip) dropped significantly when sites on the lower lip were touched by the probe. In experiment 2 (*b*), touch was applied to each of four quadrants in random order. Each time touch was applied, participants gave separate tingling ratings for each quadrant, after being prompted in random order. Data from all four touch conditions were realigned to the left upper lip position to express the spatial relation between the location where tingling intensity was judged and the location where sustained touch was applied. Significant intensity reduction was observed at the location where the touch was applied. (*d*) In experiment 3 (*n* = 8), the probability of judging the lower lip tingling intensity as stronger than the upper lip decreased as force level increased. The black line represents the sample average, the grey shading represents the s.e.m., and coloured lines represent individual data. (*e*) In experiment 4 (*n* = 14), the estimated vibration amplitude of sanshool tingling decreased as a static force increased. Moreover, the intensity of tingling significantly recovered as time elapsed. Error bars indicate the s.e.m. across participants and coloured dots indicate individual data. (Online version in colour.)

significantly from the normal distribution (see the electronic supplementary material, table S7). The statistical analysis was therefore conducted after log-transforming the data. However, to maintain the data in interpretable scale, we report and show the means and the standard errors in the original units (µm). The initial perceived tingling on the lower lip without pressure was matched by, on average, 13.9 µm (± s.d. 5.9) peak-to-peak amplitude of a 50 Hz vibration on the upper lip. Sustained contact force of 0.05 N on the lower lip reduced the tingling to a level that was now matched by 8.4 µm (± s.d. 4.6) of vibration amplitude. Contact forces of 0.20 N and 0.35 N were matched by 8.2 µm (± s.d. 4.1) and 7.6 µm

(± s.d. 3.4) vibration amplitudes, respectively (figure 2*e*). For each contact force level, the perceived intensity of tingling was significantly reduced at all time points (pressure onset, +5 s, and +10 s after pressure onset) compared to the initial baseline period without any pressure contact ($p < 0.05$ corrected for all nine comparisons), replicating the result of experiments 1–3.

We specifically wanted to investigate whether the reduction of tingling sensation would change with the force level applied, and whether that reduction would change as a function of time from onset of probe contact. A 3 (force: 0.05, 0.20 and 0.35 N) × 3 (time: force onset, +5 s, and +10 s

after force onset) repeated measures ANOVA on the vibration amplitude showed significant main effect for both factors of contact force level ($F_{2,26} = 4.32$; $p = 0.024$; $\eta_p^2 = 0.249$) and time since contact ($F_{2,26} = 4.92$; $p = 0.015$; $\eta_p^3 = 0.275$), but no significant interaction effect ($F_{4,52} = 0.27$; $p = 0.897$) (figure 2e).

We used Fisher's LSD methods to identify conditions that differed significantly. For the force level factor, estimated tingling amplitude was significantly reduced in the highest (0.35 N) compared to the middle force level (0.20 N) ($t_{13} = 3.54$, $p = 0.004$, dz = 0.94). Comparison with the lowest force level (0.05 N) showed a similar trend ($t_{13} = 2.15$, $p = 0.051$, dz = 0.57) (figure 2e). Therefore, the intensity of tingling was suppressed in a force-dependent fashion, as expected from experiment 3. We investigated the effect of time in the same way. The perceived tingle intensity recovered as time elapsed (onset versus +5 s: $t_{13} = 2.82$, $p = 0.014$, dz = 0.75; onset versus +10 s: $t_{13} = 2.38$, $p = 0.033$, dz = 0.63) (figure 2e). Because activity of SA neurons gradually reduces over time owing to the adaptation to sustained pressure input [24], this modest time-dependent recovery of tingling sensation is consistent with the hypothesis that activity of SA neurons underlies suppression of RA-mediated sanshool-tingling.

## 4. Discussion

Somatosensory perception involves integration of multiple features that reach the brain through different afferent channels. A central question is therefore whether and how inputs from these different channels interact with each other [2,14,27]. Classical theories suggested that specific frequency-selective channels, associated with specific receptors and afferent fibre types, were processed independently, at least until early sensory cortex [10]. A direct test of such independence would measure whether cortical or perceptual responses to a given frequency channel did or did not depend on whether other frequency channels were also activated. Such a test would use multi-frequency mechanical stimuli, which could lead to complex mechanical interactions within the skin and soft tissues [16,28]. The effective stimulation at the receptor during multi-frequency stimulation is therefore unknown. As a result, any given pattern of inter-channel interactions observed in neural or perceptual data could be mechanical rather than neural in origin. This makes hypotheses of independent neural frequency channels difficult to test. While some neuronal studies have begun to challenge the classical view of independent frequency channels [2,14,15,29], direct *perceptual* evidence for neural interactions between somatosensory submodalities has been lacking until now. We have used a novel approach involving anomalous chemical stimulation of mechanoreceptor channels, to show strong inhibitory interactions between distinct perceptual channels encoding different preferred frequencies. Specifically, we show that the tingling sensation associated with the flutter-range vibratory channel (putative RA channel) is inhibited by the input of sustained pressure (putative SA channel). We further showed that this inhibitory interaction is spatially selective and proportional to the activation of the pressure channel. By combining a mechanical stimulus with an anomalous chemical stimulus, we could avoid the methodological uncertainties of possible nonlinear mechanical interactions between mechanical stimuli that may affect other studies.

In the current study, we investigated perceptual channels based on psychophysically defined characteristics. These methods identify perceptual channels by threshold differences across different stimulus frequencies, and by observing perceptual modulations owing to adaptation and masking. Although the peripheral (receptor/afferent fibre) basis of tactile feature processing have been extensively studied by neurophysiologists, we still do not know the precise details of the mapping between channels defined by peripheral physiology, and the perceptual channels defined by psychophysics. Nevertheless, the principle of studying principles of central nervous system (CNS) organisation based on psychophysically defined perceptual channels has been well established, for example in the visual system [30]. By analogy to visual psychophysics studies, we believe that the tactile feature processing system can also be usefully investigated by studying interaction between perceptual channels.

Sanshool has been shown to activate the rapidly adapting light-touch fibres in rats [17,20]. Using both adaptation [21] and masking paradigms [22], previous studies have demonstrated that a flutter-range vibration channel (putative RA channel) activation is responsible for the sanshool-induced tingling sensation. First, the perceived sanshool-induced tingling frequency on the lips is reduced by adapting the RA channel using prolonged mechanical vibration [21], paralleling the reduction in perceived frequency of mechanical vibration by similar adaptation procedures. Second, application of sanshool on the skin impairs detection of 30 Hz mechanical vibration (RA channel dominant frequency) but does not affect detection of 240 Hz (Pacinain corpuscle (PC) channel dominant frequency) or 1 Hz (SA channel dominant frequency) mechanical vibration [22], demonstrating that sanshool can selectively affect the putative RA channel. Finally, microstimuluation studies confirm the strong link between RA activation and flutter-range vibration sensations [31]. Thus, although we could not directly measure RA afferent responses to sanshool, we may nevertheless study the perceptually defined channel underlying the sanshool tingling sensation, while identifying this putatively as an RA channel. Future microneurographic studies could potentially provide stronger evidence about the physiological afferents responsive to sanshool, including selectivity for particular afferent types.

Nevertheless, psychophysical techniques can also help to investigate whether other non-mechanical channels might contribute to sanshool tingling sensations (electronic supplementary material, Methods and figure S5A-B). C-nociceptive and C-tactile fibres have both been suggested to mediate tingle. Although nociceptive A$\beta$ neurons have also been recently described in humans [32], it is not clear whether these fibres are also activated by sanshool. Moreover, although C-tactile fibres are commonly found only on hairy skin [3], there is some electrophysiological [33] as well as psychophysical evidence [34,35] of the existence of C low-threshold mechanoreceptors in the glabrous skin. Thus, animal studies have shown that sanshool indeed activates small fibres [17,20] as well as RA fibres. However, we performed two control experiments, which suggested that neither C-nociceptive nor C-tactile fibres contribute to the tingling sensation. First, we found that perceived intensity of sanshool-induced tingling was unaffected by topical lidocaine anaesthetics that preferentially blocked the small fibres mediating pain sensation [36,37] (electronic

supplementary material, figure S6A-B). Second, we found that tingling sensation intensity increased linearly and monotonically as a function of stimulation temperature, in clear contrast to the inverted-U function of temperature that characterises C-tactile firing [38] (electronic supplementary material, figure S6C). The thermal sensitivity of C-tactile fibres remains controversial, with some studies finding C-tactile responses to cooling [34,39], rather than the inverted-U shape [38]. The results of our control experiment were incompatible with both of these reported patterns of C-tactile thermal modulation, yet were compatible with the reported pattern of thermal modulation of RA firing [40,41]. Our psychophysical observations also agree with evidence from microneurography [42] and clinical neuropathies [43,44], which both consistently identify tingling paraesthesias with activation of large-diameter afferents. By contrast, activation of small-diameter afferents generally elicits low, dull, painful sensations.

We found that sustained light touch attenuated sanshool tingle, and we propose that this reflects an interaction between the corresponding perceptual channels. Our experimental design successfully controlled for several alternative possible explanations of touch-induced suppression of tingling. First, we ruled out the possibility that sustained touch may have attracted attention to mechanical stimulus, either distracting attention away from the tingling sensation, or masking sanshool-induced activity in the same channel [45,46]. Explanations based on distraction cannot readily explain why suppression of tingle was location-dependent, with stronger suppression of tingling at the location of touch compared to remote from it. Alternative explanations based on masking would require the steady pressure stimulus to activate the same perceptual channel as sanshool, i.e. the putative RA channel. RA afferents typically respond at the onset of a steady pressure, but lack a sustained response [5,47]. Intra-channel masking theories would therefore predict transient suppression of tingle sensation at the onset of steady pressure, with rapid rebound of tingle during continued tactile contact. Yet, in experiments 1, 2 and 4 we found significant touch-induced attenuation of sanshool tingle after 10 s of continuous touch, suggesting that an RA contribution to the attenuation of tingle is unlikely. Moreover, in experiments 3 and 4 we found that pressure-induced attenuation of sanshool-evoked tingling increases linearly with the indentation. Linear increase in firing rate with indentation is a characteristic marker of SA fibres [24,32,48], but is absent in RA fibres, which are instead mostly affected by indentation velocity [47]. Thus, overall, the effects of steady pressure on sanshool-evoked tingling are consistent with steady pressure conveyed by a putative SA pathway, influencing sanshool-induced activity in a putative RA pathway.

Another alternative explanation is based on the effective stimulation at the receptors themselves. Recent studies showed that action potentials are accompanied by mechanical deformations of the cell surface [49,50], as well as mechanical waves propagating throughout the axonal surface [51]. Therefore, one possibility is that sustained pressure might have changed the neural response of RA mechanoreceptors or their afferent fibres to sanshool, as a secondary consequence of physically deforming their shape. However, the spatial tuning pattern of our effects offer evidence against this hypothesis. In experiment 2, sustained touch-related tingling inhibition was strongest at the place of the touch

stimulation itself, and at other locations on the same lip. However, we also found that the tingling sensation on the upper lip was modulated by touch on the lower lip and vice versa. Because the lips were held apart during the experiment, this modulation cannot readily be explained by the spread of mechanical input across the lips. Moreover, upper and lower lip are innervated by different branches of the trigeminal nerve (V2 and V3, respectively), which would not allow purely peripheral interactions. Finally, the time-course of suppression is inconsistent with a direct effect of sustained pressure on the RA receptor or itself, or its afferent. In experiment 4, tingling levels were strongly suppressed immediately after static touch was applied, but then recovered significantly over the subsequent 10 s (figure 2e). A direct mechanical effect on the RA receptor should presumably remain constant as long as sustained touch lasts. By contrast, the modest recovery of tingle with continuing pressure is consistent with a neural, as opposed to mechanical account, based on the adaptation of SA afferent firing rates.

Our study presents a series of limitations, which should be studied in more detail in the future. First, in our study, channels are defined *perceptually*, and their identification with specific peripheral receptors and afferent fibres can only be putative. Although the physiological characteristics of sanshool are well studied in animal research [17–20], the physiological profile of peripheral mechanoreceptive activation induced by sanshool in humans has not yet been investigated directly, and is known only by psychophysical proxy measures. Future studies could potentially record single peripheral afferents from the human skin microneurographically, and identify the response of different fibre classes to sanshool applied to their respective receptive fields.

Second, the perceptual characteristics of sanshool tingling should be studied in more detail. In our study, we focus on the feature of flutter-level vibration, but other aspects of the sensation remain to be systematically investigated. For example, in a previous study, we have shown that sanshool produces tingling at a frequency of 50 Hz [21] and impairs detection of mechanical vibrations at 30 Hz but not higher (240 Hz) or lower (1 Hz) frequencies [22]. The present study extends knowledge of sanshool's sensory properties by confirming that the perceived intensity of sanshool tingling is dose-dependent (experiment 3) [52].

Third, the duration of static touch varied widely across our experiments (10 s in experiments 1, 2 and 4, 1 s in experiment 3). We varied the duration of static touch because our experiments required different numbers of tactile stimulation trials, which had all to be completed within the typical duration of tingling that follows a single application of sanshool (approx. 40 min). Despite the varying tactile durations, we consistently found suppression of tingling sensation, suggesting a rather general effect.

Finally, some of our experiments involved manual delivery of tactile stimuli. These cannot provide precise control over contact force. Given that RA mechanoreceptors are exquisitely sensitive to dynamic changes in contact force, our experiments 1 and 2 may have included uncontrolled micromovements that activated RA channels. Nevertheless, the precisely controlled mechanical stimuli of experiments 3 and 4, which should have drastically reduced micromovements, also produced a strong attenuation of tingling, several seconds after touch onset. These results suggest that the tactile attenuation of tingling is likely to be mediated by

an SA rather than by an RA channel activated by unintended micromovements.

At what level in the CNS, then, would putative RA and SA channels interact? Either cortical or sub-cortical interactions are possible. Several circuit mechanisms of presynaptic inhibition have recently been described [53]. In the mouse spinal cord, different types of interneurons in the dorsal horn receive inputs from multiple types of low threshold mechanoreceptor afferents (including RA and SA channels). Because the inhibition of pain by touch (SA channel) is thought to occur at the dorsal horn [6,54,55], the mechanism of analogous inhibition of RA activity by SA input might also be implemented sub-cortically, e.g. spinally or in trigeminal nuclei for somatic or orofacial stimuli respectively. Within somatosensory cortex, neurons in each frequency channel were originally thought to be organised in discrete functional columns [10]. However, many single neurons in area 3b/1 show hybrid activity profiles responding to both RA (transient) and SA (sustained activity) mechanical input [11,15,56]. Therefore, interactions between sub-modalities may occur prior to somatosensory cortex [29].

What might be the functional relevance of a putative SA-RA mechanoreceptor channel interaction? A few studies have previously investigated whether vibration perception is affected by the indentation of the vibrotactile stimulator [57,58]. For example, Lowenthal et al. [57] found that detection thresholds for vibratory stimuli are significantly lower at higher contact force levels. However, although seemingly inconsistent with our finding that pressure inhibits vibration perception, Lowenthal's results may be owing to physical interactions between the mechanical stimuli, rather than neural interactions between the resulting signals. More generally, studies with complex mechanical stimuli cannot readily rule out the possibility that apparent interactions between different frequency-tuned tactile channels are in fact owing to nonlinear mechanical interactions in the periphery, which influence the effective stimulation at the receptor. By contrast, by using sanshool as an anomalous chemical stimulus for cutaneous receptors, we were able to reliably deliver tingling sensations in the absence of any mechanical confounds (e.g. the pressure exerted by the probe of the vibrotactile stimulator).

To our knowledge, the current study is the first to suggest an inhibitory effect of SA on RA signalling. However, previous reports of an effect in the *reverse* direction, from RA to SA signalling, offer important clues to possible function of such interactions. Bensmaia and colleagues [59,60]

showed that increasing the ratio of RA firing to SA firing impaired grating detection performance: RA input interfered with perception of fine spatial structure carried by SA. We, therefore, speculate that the tactile system contains mechanisms to inhibit RA channel input, to prevent masking by RA-mediated noise, and in order to maintain the robustness and stability of tactile perception. For example, when any tactile contact occurs, mechanical waves [16,61] travel through the skin, and deeper tissues. Interestingly, RA-range frequencies travel over considerable distances. We speculate that SA-induced suppression of RA firing, as reported here, could play an important role in limiting the perceptual impact of these complex mechanical interactions. Lateral inhibition between neurons with adjacent receptive fields is a pervasive feature of sensory spatial representation, serving to increase spatial acuity [62,63]. Lateral inhibition occurs also for non-spatial sensory systems, such as olfaction, where it again serves to enhance perceptual resolution. Our findings are consistent with a functional hypothesis that inhibition of one frequency channel by another frequency channel functions analogously to the enhancement of spatial acuity provided by lateral inhibition. SA-mediated suppression of RA activation during normal touch may serve as a low-pass filter mechanism, allowing reliable perception of tactile events at sensorimotor timescales.

**Ethics.** All participants gave their informed written consent. Experiments 1–5 were approved by University College London Research Ethics Committee. Experiment 6 was approved by the Research Ethics Committee of the School of Advanced Study, University of London.

**Data accessibility.** All the data used for the statistical inferences of the paper are available in the electronic supplementary material.

**Authors' contribution.** A.C., N.H. and P.H. conceived the study. N.H. and P.H. designed, and Y.H. and N.H. collected and analysed the data for experiments 1 and 2. A.C. and P.H. designed, and A.C. collected and analysed the data for experiments 3–6. A.C. produced the figures. A.C., N.H. and P.H. wrote the paper. All the authors reviewed the paper.

**Competing interests.** The authors declare no competing interests.

**Funding.** This research was supported by an MRC project grant no. MR/M013901/1 to P.H. A.C. and P.H. were further supported by a donation by Dr Shamil Chandaria to the Institute of Philosophy, School of Advanced Study, University of London. N.H. is supported by Japan Society for the Promotion of Science (Kakenhi 26119535, 18H01106) and ERATO (JPMJER1801). Y.H. is supported by The Great Britain Sasakawa Foundation.

**Acknowledgements.** We are grateful to Indena SPA (Milan, Italy) for providing ZANTHALENE used for the study.

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
