## [Reviewer comments · Proceedings of the Royal Society B: Biological Sciences]

Review History

RSPB-2020-0717.R0 (Original submission)

Review form: Reviewer 1

Recommendation

Reject – article is scientifically unsound

Scientific importance: Is the manuscript an original and important contribution to its field?

Marginal

General interest: Is the paper of sufficient general interest?

Good

Quality of the paper: Is the overall quality of the paper suitable?

Marginal

Is the length of the paper justified?

Yes

Should the paper be seen by a specialist statistical reviewer?

No

Do you have any concerns about statistical analyses in this paper? If so, please specify them explicitly in your report.

No

It is a condition of publication that authors make their supporting data, code and materials available - either as supplementary material or hosted in an external repository. Please rate, if applicable, the supporting data on the following criteria.

Is it accessible?

Yes

Is it clear?

Yes

Is it adequate?

Yes

Do you have any ethical concerns with this paper?

No

Comments to the Author

In the present study, Cataldo and colleagues characterized the effect of sustained pressure, temperature, and anesthesia on the tingling intensity evoked by the activation of the rapidly adapting mechanoreceptors (RA) using hydroxy-alpha-sanshool (sanshool). This experimental approach seems suitable to test the interactions across somatosensory sub-modalities, but there are important caveats that limit the conclusions. In particular, the authors claim to manipulate selectively the slowly adapting mechanoreceptors (SA) system using mechanical stimulation. However, there is no physiological evidence in the current study to confirm that the constant pressure stimuli used only activate the SA system. Because of this limitation, any conclusion the authors draw on the perception of the sanshool stimulus needs to be based on how the RA channel responded to the pairing of the mechanical stimulation and sanshool – the authors cannot definitively tease apart the influences of the mechanical test stimuli on the RA and SA systems using only their psychophysical methods. Unfortunately, the novelty of the authors' study rests on the claim of across-channel interactions, which cannot be conclusively demonstrated without neurophysiological evidence. Accordingly, we cannot recommend publication. We provide more detailed comments below that we hope can be helpful for improving future versions of the manuscript.

1) The results from experiments 1-4 demonstrate that the perceived tingling intensity is affected by sanshool concentration, differences in mechanical pressure, and when intensity judgements are performed (relative to the mechanical pressure onset). However, there was no systematic evaluation of the dose-time-effect on the tingling perception. The best approach seems to be the experiment 4 design (tingling intensity match) but, why did the authors not characterize the frequency and amplitude that best match the tingling sensation as a function of sanshool dose/time? This step is crucial to describe the psychophysical attributes of the perception evoked by sanshool. It will help to characterize which attributes (frequency or amplitude) are affected when other stimuli are presented. And even more, it will shed light on the perceptual variability across subjects. For example, which is the best frequency and amplitude to describe the 20% sanshool effect? How reliable/variable is the effect as a function of time (i.e. there is a plateau or a peak intensity effect)? Does the frequency or the amplitude (or both) change as a function of dose and time? Is 50 Hz the best frequency to match the tingling intensity across subjects? The absence of calibration limits the interpretation of the authors' results.

2) The results from experiment 4 showed that the most prominent effect of tactile pressure on tingling intensity occurred immediately after the stimulus started. This period also happens to be

the interval during which the RA and SA channels are likely to be co-activated by the mechanical stimulation. Therefore, in the absence of neurophysiological evidence, it is important to be cautious about claiming that only one group of mechanoreceptors is responsible for the perceptual effect. For example, from experiment 1 & 2, how much of the results were biased by the early part of the tactile stimulation?

3) Related to figure 3, why did the authors not include the amplitude before the tactile pressure? Panel B and C must show the 10 conditions (3 pressure X 3 times from onset and 1 amplitude before tactile stimulation). Also, the authors must present the data from individual subjects in all of the experiments rather than simply providing the group-averaged summary statistics. This standard practice is important for the reader to appreciate the across-subject variability/consistency. This is particularly important given the relatively low number of stimulus repetitions for particular conditions tested in some experiments.

4) With exception of experiments 1 and 2, either the mechanical probe or the high/low dose location was fixed across trials and subjects. Is there any reason that the test tactile/temperature/anesthesia stimuli were always applied to the lower lip? With such a regular and consistent stimulus, it is not obvious that the subject even needs to attend to the other stimulus in the two-interval design. It would have been better to randomize the delivery of the mechanical stimulus. Furthermore, if the dose location cannot be randomized on a trial by trial basis, at least should be done across subjects. Also, it is important to explain why different stimulation durations were used across experiments (exp 1,2 & 4 = 10 s, exp 3 = 1 s, exp 6 = 4 s)?

5) What is the density of C-tactile fibers on the lips? The authors cite evidence that these fibers are found in the hairy skin (page 6; "In particular, C-tactile fibers are only found in hairy, but not glabrous skin"), but appear to base their conclusions on the effects of temperature manipulations on an assumption that C-fibers are also found on the lips. The assumptions of experiment 6 must be re-evaluated and clarified. Also, it is important to report the pressure applied in this experiment.

Review form: Reviewer 2

Recommendation

Accept as is

Scientific importance: Is the manuscript an original and important contribution to its field?

Excellent

General interest: Is the paper of sufficient general interest?

Excellent

Quality of the paper: Is the overall quality of the paper suitable?

Excellent

Is the length of the paper justified?

Yes

Should the paper be seen by a specialist statistical reviewer?

No

Do you have any concerns about statistical analyses in this paper? If so, please specify them explicitly in your report.

No

It is a condition of publication that authors make their supporting data, code and materials available - either as supplementary material or hosted in an external repository. Please rate, if applicable, the supporting data on the following criteria.

Is it accessible?

Yes

Is it clear?

Yes

Is it adequate?

Yes

Do you have any ethical concerns with this paper?

No

Comments to the Author

This paper is a joy to read. Original; informative; clearly written; with a logic sequence of experiments covering the relevant question in an adequate and ingenious way.

“Chapeau...”

Review form: Reviewer 3

Recommendation

Accept with minor revision (please list in comments)

Scientific importance: Is the manuscript an original and important contribution to its field?

Good

General interest: Is the paper of sufficient general interest?

Good

Quality of the paper: Is the overall quality of the paper suitable?

Good

Is the length of the paper justified?

Yes

Should the paper be seen by a specialist statistical reviewer?

Yes

Do you have any concerns about statistical analyses in this paper? If so, please specify them explicitly in your report.

No

It is a condition of publication that authors make their supporting data, code and materials available - either as supplementary material or hosted in an external repository. Please rate, if applicable, the supporting data on the following criteria.

Is it accessible?

No

Is it clear?

No

Is it adequate?

No

Do you have any ethical concerns with this paper?

No

Comments to the Author

This is an interesting and important psychophysical study showing the inhibition of chemically evoked tingling by pressure.

Main concern

To my knowledge, there are no direct nerve recordings (microneurography) in humans with sanshool. If the authors have electrophysiological data in humans showing that sanshool activates RA1 afferents, and does so selectively, that will really strengthen this paper. The finding that “touch inhibits touch” is interesting in its own right, but in the absence of electrophysiological data, I am not convinced with the link drawn between stimulus/percept and the afferent type. Let’s take the RAs first. If this presumed link is based on intraneural microstimulation studies, these have shown that the percept evoked by activation of RA afferents is frequency-dependent, and at higher stimulation frequencies, the percept evoked by activation of RA1 or RA2/PC afferent is qualitatively indiscernible (Ochoa and Torebjörk 1983, JPhysiol). Most microstimulation studies were performed on the glabrous skin of the hand, so whether those observations can be extrapolated to other body regions, like the lip, in this case, is unclear. Also, the idea that the RA1s alone are responsible for encoding flutter-range frequency has been challenged in a recent paper showing that the PCs can do that just as well (Birznieks et al. 2019, eLife), indicating that it is the spiking pattern, and not the receptor type, that determines frequency perception. The same argument applies to sustained indentation. The authors refer to sustained indentation as SA1-mediated, but SA1s are not the only afferent type that responds to sustained indentation. SA2s obviously have a sustained response, and at higher indentation forces, even field afferents show some sustained activity (Nagi et al. 2019, Sci Adv). Also, CTs have some sustained response.

Other comments/suggestions:

- Vibration thresholds decrease as the pressure of applied stimulus is increased (e.g. Lowenthal et al. 1987, Diabetes Care). This should be discussed in the context of the findings of the current study.
- Weerakkody et al. (2007, JPhysiol) showed an impairment of proprioceptive acuity when high-frequency vibration was applied but not when low-frequency vibration was applied. Proprioception has a cutaneous component, some even consider it to be part of the tactile domain, and these interactions I think are relevant and should be acknowledged here.
- “Experiment 1 the lips were manually stimulated by the experimenter with a probe (diameter: 14 mm, force: ~1.5 N) for 10 seconds.” This is handheld without force-feedback control – RAs, for instance, are exquisitely sensitive to dynamic changes, and any small movement that may or may not be perceptible could make the RAs fire.
- Please provide the corresponding value in pressure for all mechanical stimulations.
- “Participants rated the tingling sensation relative to its intensity during the no-pressure baseline: 0 as no sensation, 10 as same level as the intensity during baseline, and rating above 10 as higher intensity than the baseline period.” Can you please clarify this?
- “The rating occurred after 10 s of continuous probe touch to avoid any mechanical transient effect.” Please refer to my comment above about micro-movements. It is handheld

stimulation so almost impossible to avoid these.

- “Each trial started when the participant confirmed that the tingling sensation had returned, typically after a few seconds.” This is quite interesting that the inhibition outlasts the stimulus by a few seconds. What does this depend on?
- I would suggest including a methods figure. There are six experiments, and I think a schematic representation of these would be helpful.
- “The experiment consisted of six blocks where static touch was applied twice to central positions 3 and 6 and once to the other areas (ten ratings from each participant per session).” Why the difference? What was the rationale for that?
- Referral to the figures does not follow the sequence in the text, for example, Fig 1 A, then 1 C...
- “Yet, instead of only rating the sanshool tingling at a fixed location, participants rated the tingling intensity for all four lip quadrants.” Can you explain this more? Why were the instructions changed in Experiment 2?
- “Forces were applied by a closed-loop system composed by a linear actuator (ZABER, XYZ Series, Vancouver, Canada) and a force gauge (McMesin, PFI-200N GEB, Slinfold, UK) (Figure 2), which continuously maintained the desired pressure level.” Please provide an example (force trace).
- “To rule out the contribution of nociceptive C-fibres to sanshool tingling, we measured both pain thresholds and perceived intensity of tingling before and after blocking the activity of small fibres through lidocaine.” How was the contribution of A-beta nociceptors (Nagi et al. 2019, *Sci Adv*) ruled out?
- “In particular, C-tactile fibres are only found in hairy, but not glabrous skin”. I would suggest revising this to say that there is no evidence of a phenotypically identical class of hairy-skin C-tactile fibers in human glabrous skin. Just to clarify my point: there is electrophysiological evidence of the existence of C low-threshold mechanoreceptors in the glabrous skin of rat hind paw (Djoughri 2016, *Neurosci Lett*), and psychophysical evidence, using nerve blocks, of a C low-threshold input from the glabrous skin of human hand (Nagi and Mahns, 2013, *Exp Br Res*; Nagi et al. 2015, *BMC Neurosci*).
- “Importantly, C-tactile fibres also show preference for neutral, skin temperature (32 °C) stimuli, rather than warm (40 °C) or cold (18 °C) stimuli [31]. Thus, if the sanshool tingle is mediated by C-tactile fibres, the perceived intensity of tingling should then be maximal at neutral temperatures and reduced during cold or warm stimulation (i.e. inverted U-shape).” It’s a very subtle effect of temperature on CT brush responsiveness, and the inverted U is still produced.
- “Instead, the interaction appears to occur at some neural processing level where afferents from the two mechanoreceptors are integrated in a spatially organised manner.” For circuit mechanisms of presynaptic inhibition have a look at the paper by Zimmerman et al. 2019 (*Neuron*).
- For the bar charts, can you please include/ superimpose the individual data points as well?
- “Cold touch produced the lowest ratings (1.85 ± SD 1.7).” There is some evidence that CTs respond to cooling e.g. Nordin 1990 (*J Physiol*), Samour et al. 2015 (*Pain*). So could an alternative interpretation be that when CTs are co-stimulated with pressure and cold, the inhibition of tingling is most robust? Of course, the same argument applies to SAs in general

which the authors make for type 1 SA, and these were indeed referred to as spurious thermoreceptors by Iggo.

- “This suppression pattern clearly differs to the inverted U-shape that would be expected from C-tactile fibre thermal sensitivity.” Can you clarify this? The inverted U is not for static touch, it’s actually not known.
- “although we cannot directly measure RA afferent responses to sanshool, we may nevertheless use sanshool tingling sensation as a proxy for RA activation.” OK good that it’s mentioned here. My point is the same – if there is no direct evidence why go there? The story is interesting even without it.
- In terms of the quality of the sensation, how similar is sanshool to low-frequency vibration?
- “Top-down filtering accounts require the implausible assumption that cold touch is more salient than warm touch to explain this result.” Why is this implausible?
- In Experiment 2, the modulation is not spatially constrained. What are the limits on that? Can touch on the forearm inhibit tingling sensation on the lip? Like it has been shown for vibration in TMJ (Fillingim et al. 1998, Pain) although that was in the context of gain of function (allodynia). On that note actually, there’s no data for other body parts for sanshool shown here. Has it been done in an earlier study? If so, how similar is the sensation to the lip? I think this question is important for the generalizability of these results.

Decision letter (RSPB-2020-0717.R0)

12-Jun-2020

Dear Dr Cataldo:

I am writing to inform you that your manuscript RSPB-2020-0717 entitled "Anomalous mechanoreceptor activation by chemical stimulation reveals novel interaction between somatosensory submodalities" has, in its current form, been rejected for publication in Proceedings B.

This action has been taken on the advice of referees, who have recommended that substantial revisions are necessary. With this in mind we would be happy to consider a resubmission, provided the comments of the referees are fully addressed. However please note that this is not a provisional acceptance.

- 1) A ‘response to referees’ document including details of how you have responded to the comments, and the adjustments you have made.

- 2) A clean copy of the manuscript and one with 'tracked changes' indicating your 'response to referees' comments document.
- 3) Line numbers in your main document.

Sincerely,
Dr Sasha Dall
mailto:proceedingsb@royalsociety.org

Associate Editor
Board Member: 1

Comments to Author:

Your manuscript has been reviewed by three experts. Although there is considerable enthusiasm for your study, several concerns have been raised. The most serious is that you do not present electrophysiological data showing that sanshool activates RA1 afferents, and does so selectively. This is a significant problem. I am recommending rejection but I would encourage you to resubmit a revised manuscript that addresses this issue.

Reviewer(s)' Comments to Author:

Referee: 1

Comments to the Author(s)

In the present study, Cataldo and colleagues characterized the effect of sustained pressure, temperature, and anesthesia on the tingling intensity evoked by the activation of the rapidly adapting mechanoreceptors (RA) using hydroxy-alpha-sanshool (sanshool). This experimental approach seems suitable to test the interactions across somatosensory sub-modalities, but there are important caveats that limit the conclusions. In particular, the authors claim to manipulate selectively the slowly adapting mechanoreceptors (SA) system using mechanical stimulation. However, there is no physiological evidence in the current study to confirm that the constant pressure stimuli used only activate the SA system. Because of this limitation, any conclusion the authors draw on the perception of the sanshool stimulus needs to be based on how the RA channel responded to the pairing of the mechanical stimulation and sanshool – the authors cannot definitively tease apart the influences of the mechanical test stimuli on the RA and SA systems using only their psychophysical methods. Unfortunately, the novelty of the authors' study rests on the claim of across-channel interactions, which cannot be conclusively demonstrated without neurophysiological evidence. Accordingly, we cannot recommend publication. We provide more detailed comments below that we hope can be helpful for improving future versions of the manuscript.

1) The results from experiments 1-4 demonstrate that the perceived tingling intensity is affected by sanshool concentration, differences in mechanical pressure, and when intensity judgements are performed (relative to the mechanical pressure onset). However, there was no systematic evaluation of the dose-time-effect on the tingling perception. The best approach seems to be the experiment 4 design (tingling intensity match) but, why did the authors not characterize the frequency and amplitude that best match the tingling sensation as a function of sanshool dose/time? This step is crucial to describe the psychophysical attributes of the perception evoked by sanshool. It will help to characterize which attributes (frequency or amplitude) are affected when other stimuli are presented. And even more, it will shed light on the perceptual variability across subjects. For example, which is the best frequency and amplitude to describe the 20%

sanshool effect? How reliable/variable is the effect as a function of time (i.e. there is a plateau or a peak intensity effect)? Does the frequency or the amplitude (or both) change as a function of dose and time? Is 50 Hz the best frequency to match the tingling intensity across subjects? The absence of calibration limits the interpretation of the authors' results.

2) The results from experiment 4 showed that the most prominent effect of tactile pressure on tingling intensity occurred immediately after the stimulus started. This period also happens to be the interval during which the RA and SA channels are likely to be co-activated by the mechanical stimulation. Therefore, in the absence of neurophysiological evidence, it is important to be cautious about claiming that only one group of mechanoreceptors is responsible for the perceptual effect. For example, from experiment 1 & 2, how much of the results were biased by the early part of the tactile stimulation?

3) Related to figure 3, why did the authors not include the amplitude before the tactile pressure? Panel B and C must show the 10 conditions (3 pressure X 3 times from onset and 1 amplitude before tactile stimulation). Also, the authors must present the data from individual subjects in all of the experiments rather than simply providing the group-averaged summary statistics. This standard practice is important for the reader to appreciate the across-subject variability/consistency. This is particularly important given the relatively low number of stimulus repetitions for particular conditions tested in some experiments.

4) With exception of experiments 1 and 2, either the mechanical probe or the high/low dose location was fixed across trials and subjects. Is there any reason that the test tactile/temperature/anesthesia stimuli were always applied to the lower lip? With such a regular and consistent stimulus, it is not obvious that the subject even needs to attend to the other stimulus in the two-interval design. It would have been better to randomize the delivery of the mechanical stimulus. Furthermore, if the dose location cannot be randomized on a trial by trial basis, at least should be done across subjects. Also, it is important to explain why different stimulation durations were used across experiments (exp 1,2 & 4 = 10 s, exp 3 = 1s, exp 6 = 4 s)?

5) What is the density of C-tactile fibers on the lips? The authors cite evidence that these fibers are found in the hairy skin (page 6; "In particular, C-tactile fibers are only found in hairy, but not glabrous skin"), but appear to base their conclusions on the effects of temperature manipulations on an assumption that C-fibers are also found on the lips. The assumptions of experiment 6 must be re-evaluated and clarified. Also, it is important to report the pressure applied in this experiment.

Referee: 2

Comments to the Author(s)

This paper is a joy to read. Original; informative; clearly written; with a logic sequence of experiments covering the relevant question in an adequate and ingenious way.

"Chapeau..."

Referee: 3

Comments to the Author(s)

This is an interesting and important psychophysical study showing the inhibition of chemically evoked tingling by pressure.

Main concern

To my knowledge, there are no direct nerve recordings (microneurography) in humans with sanshool. If the authors have electrophysiological data in humans showing that sanshool activates RA1 afferents, and does so selectively, that will really strengthen this paper. The finding

that “touch inhibits touch” is interesting in its own right, but in the absence of electrophysiological data, I am not convinced with the link drawn between stimulus / percept and the afferent type. Let’s take the RAs first. If this presumed link is based on intraneural microstimulation studies, these have shown that the percept evoked by activation of RA afferents is frequency-dependent, and at higher stimulation frequencies, the percept evoked by activation of RA1 or RA2/PC afferent is qualitatively indiscernible (Ochoa and Torebjörk 1983, *J Physiol*). Most microstimulation studies were performed on the glabrous skin of the hand, so whether those observations can be extrapolated to other body regions, like the lip, in this case, is unclear. Also, the idea that the RA1s alone are responsible for encoding flutter-range frequency has been challenged in a recent paper showing that the PCs can do that just as well (Birznieks et al. 2019, *eLife*), indicating that it is the spiking pattern, and not the receptor type, that determines frequency perception. The same argument applies to sustained indentation. The authors refer to sustained indentation as SA1-mediated, but SA1s are not the only afferent type that responds to sustained indentation. SA2s obviously have a sustained response, and at higher indentation forces, even field afferents show some sustained activity (Nagi et al. 2019, *Sci Adv*). Also, CTs have some sustained response.

Other comments/ suggestions:

- Vibration thresholds decrease as the pressure of applied stimulus is increased (e.g. Lowenthal et al. 1987, *Diabetes Care*). This should be discussed in the context of the findings of the current study.
- Weerakkody et al. (2007, *J Physiol*) showed an impairment of proprioceptive acuity when high-frequency vibration was applied but not when low-frequency vibration was applied. Proprioception has a cutaneous component, some even consider it to be part of the tactile domain, and these interactions I think are relevant and should be acknowledged here.
- “Experiment 1 the lips were manually stimulated by the experimenter with a probe (diameter: 14 mm, force: ~1.5 N) for 10 seconds.” This is handheld without force-feedback control – RAs, for instance, are exquisitely sensitive to dynamic changes, and any small movement that may or may not be perceptible could make the RAs fire.
- Please provide the corresponding value in pressure for all mechanical stimulations.
- “Participants rated the tingling sensation relative to its intensity during the no-pressure baseline: 0 as no sensation, 10 as same level as the intensity during baseline, and rating above 10 as higher intensity than the baseline period.” Can you please clarify this?
- “The rating occurred after 10 s of continuous probe touch to avoid any mechanical transient effect.” Please refer to my comment above about micro-movements. It is handheld stimulation so almost impossible to avoid these.
- “Each trial started when the participant confirmed that the tingling sensation had returned, typically after a few seconds.” This is quite interesting that the inhibition outlasts the stimulus by a few seconds. What does this depend on?
- I would suggest including a methods figure. There are six experiments, and I think a schematic representation of these would be helpful.
- “The experiment consisted of six blocks where static touch was applied twice to central positions 3 and 6 and once to the other areas (ten ratings from each participant per session).” Why the difference? What was the rationale for that?
- Referral to the figures does not follow the sequence in the text, for example, Fig 1 A, then 1 C...

- “Yet, instead of only rating the sanshool tingling at a fixed location, participants rated the tingling intensity for all four lip quadrants.” Can you explain this more? Why were the instructions changed in Experiment 2?
- “Forces were applied by a closed-loop system composed by a linear actuator (ZABER, XYZ Series, Vancouver, Canada) and a force gauge (McMesin, PFI-200N GEB, Slinfold, UK) (Figure 2), which continuously maintained the desired pressure level.” Please provide an example (force trace).
- “To rule out the contribution of nociceptive C-fibres to sanshool tingling, we measured both pain thresholds and perceived intensity of tingling before and after blocking the activity of small fibres through lidocaine.” How was the contribution of A-beta nociceptors (Nagi et al. 2019, Sci Adv) ruled out?
- “In particular, C-tactile fibres are only found in hairy, but not glabrous skin”. I would suggest revising this to say that there is no evidence of a phenotypically identical class of hairy-skin C-tactile fibers in human glabrous skin. Just to clarify my point: there is electrophysiological evidence of the existence of C low-threshold mechanoreceptors in the glabrous skin of rat hind paw (Djouhri 2016, Neurosci Lett), and psychophysical evidence, using nerve blocks, of a C low-threshold input from the glabrous skin of human hand (Nagi and Mahns, 2013, Exp Br Res; Nagi et al. 2015, BMC Neurosci).
- “Importantly, C-tactile fibres also show preference for neutral, skin temperature (32 °C) stimuli, rather than warm (40 °C) or cold (18 °C) stimuli [31]. Thus, if the sanshool tingle is mediated by C-tactile fibres, the perceived intensity of tingling should then be maximal at neutral temperatures and reduced during cold or warm stimulation (i.e. inverted U-shape).” It’s a very subtle effect of temperature on CT brush responsiveness, and the inverted U is still produced.
- “Instead, the interaction appears to occur at some neural processing level where afferents from the two mechanoreceptors are integrated in a spatially organised manner.” For circuit mechanisms of presynaptic inhibition have a look at the paper by Zimmerman et al. 2019 (Neuron).
- For the bar charts, can you please include/superimpose the individual data points as well?
- “Cold touch produced the lowest ratings ($1.85 \pm SD 1.7$).” There is some evidence that CTs respond to cooling e.g. Nordin 1990 (J Physiol), Samour et al. 2015 (Pain). So could an alternative interpretation be that when CTs are co-stimulated with pressure and cold, the inhibition of tingling is most robust? Of course, the same argument applies to SAs in general which the authors make for type 1 SA, and these were indeed referred to as spurious thermoreceptors by Iggo.
- “This suppression pattern clearly differs to the inverted U-shape that would be expected from C-tactile fibre thermal sensitivity.” Can you clarify this? The inverted U is not for static touch, it’s actually not known.
- “although we cannot directly measure RA afferent responses to sanshool, we may nevertheless use sanshool tingling sensation as a proxy for RA activation.” OK good that it’s mentioned here. My point is the same – if there is no direct evidence why go there? The story is interesting even without it.
- In terms of the quality of the sensation, how similar is sanshool to low-frequency vibration?
- “Top-down filtering accounts require the implausible assumption that cold touch is more salient than warm touch to explain this result.” Why is this implausible?

- In Experiment 2, the modulation is not spatially constrained. What are the limits on that? Can touch on the forearm inhibit tingling sensation on the lip? Like it has been shown for vibration in TMJ (Fillingim et al. 1998, Pain) although that was in the context of gain of function (allodynia). On that note actually, there's no data for other body parts for sanshool shown here. Has it been done in an earlier study? If so, how similar is the sensation to the lip? I think this question is important for the generalizability of these results.

Author's Response to Decision Letter for (RSPB-2020-0717.R0)

See Appendix A.

RSPB-2020-2914.R0

Review form: Reviewer 1

Recommendation

Accept as is

Scientific importance: Is the manuscript an original and important contribution to its field?

Good

General interest: Is the paper of sufficient general interest?

Good

Quality of the paper: Is the overall quality of the paper suitable?

Acceptable

Is the length of the paper justified?

Yes

Should the paper be seen by a specialist statistical reviewer?

No

Do you have any concerns about statistical analyses in this paper? If so, please specify them explicitly in your report.

No

It is a condition of publication that authors make their supporting data, code and materials available - either as supplementary material or hosted in an external repository. Please rate, if applicable, the supporting data on the following criteria.

Is it accessible?

N/A

Is it clear?

N/A

Is it adequate?

N/A

Do you have any ethical concerns with this paper?

No

Comments to the Author

The authors have addressed our major comments. The extensive revisions to the manuscript improve the data presentation, clarify the limitations of the study, and appropriately frame the results interpretation.

Decision letter (RSPB-2020-2914.R0)

18-Dec-2020

Dear Dr Cataldo

I am pleased to inform you that your Review manuscript RSPB-2020-2914 entitled "Touch inhibits touch: sanshool-induced paradoxical tingling reveals perceptual interference between somatosensory submodalities" has been accepted for publication in Proceedings B.

The referee(s) do not recommend any further changes. Therefore, please proof-read your manuscript carefully and upload your final files for publication. Because the schedule for publication is very tight, it is a condition of publication that you submit the revised version of your manuscript within 7 days. If you do not think you will be able to meet this date please let me know immediately.

To upload your manuscript, log into <http://mc.manuscriptcentral.com/prsb> and enter your Author Centre, where you will find your manuscript title listed under "Manuscripts with Decisions." Under "Actions," click on "Create a Revision." Your manuscript number has been appended to denote a revision.

You will be unable to make your revisions on the originally submitted version of the manuscript. Instead, upload a new version through your Author Centre.

- 1) A text file of the manuscript (doc, txt, rtf or tex), including the references, tables (including captions) and figure captions. Please remove any tracked changes from the text before submission. PDF files are not an accepted format for the "Main Document".
- 2) A separate electronic file of each figure (tiff, EPS or print-quality PDF preferred). The format should be produced directly from original creation package, or original software format. Please note that PowerPoint files are not accepted.
- 3) Electronic supplementary material: this should be contained in a separate file from the main text and the file name should contain the author's name and journal name, e.g `authorname_procb_ESM_figures.pdf`
All supplementary materials accompanying an accepted article will be treated as in their final form. They will be published alongside the paper on the journal website and posted on the online figshare repository. Files on figshare will be made available approximately one week before the accompanying article so that the supplementary material can be attributed a unique DOI. Please see: <https://royalsociety.org/journals/authors/author-guidelines/>
- 4) Data-Sharing and data citation

It is a condition of publication that data supporting your paper are made available. Data should be made available either in the electronic supplementary material or through an appropriate repository. Details of how to access data should be included in your paper. Please see <https://royalsociety.org/journals/ethics-policies/data-sharing-mining/> for more details.

If you wish to submit your data to Dryad (<http://datadryad.org/>) and have not already done so you can submit your data via this link <http://datadryad.org/submit?journalID=RSPB&manu=RSPB-2020-2914> which will take you to your unique entry in the Dryad repository.

Once again, thank you for submitting your manuscript to Proceedings B and I look forward to receiving your final version. If you have any questions at all, please do not hesitate to get in touch.

Sincerely,

Dr Sasha Dall
<mailto:proceedingsb@royalsociety.org>

Associate Editor
 Board Member
 Comments to Author:

I am pleased to recommend acceptance of your manuscript. You and your colleagues did an excellent job in responding to my comments and those of the reviewers. Congratulations on a fine paper.

Reviewer(s)' Comments to Author:

Referee: 1

Comments to the Author(s).

The authors have addressed our major comments. The extensive revisions to the manuscript improve the data presentation, clarify the limitations of the study, and appropriately frame the results interpretation.

Decision letter (RSPB-2020-2914.R1)

23-Dec-2020

Dear Dr Cataldo

I am pleased to inform you that your manuscript entitled "Touch inhibits touch: sanshool-induced paradoxical tingling reveals perceptual interference between somatosensory submodalities" has been accepted for publication in Proceedings B.

Open Access

Paper charges

Sincerely,

Proceedings B

Appendix A

RSPB-2020-0717

Touch inhibits touch: sanshool-induced paradoxical tingling reveals perceptual interference between somatosensory submodalities

London, 30th of November 2020

Dear Editor,

Thank you very much for requesting a revised version of our manuscript previously entitled “Anomalous mechanoreceptor activation by chemical stimulation reveals novel interaction between somatosensory submodalities”, which now has a new title: “Touch inhibits touch: sanshool-induced paradoxical tingling reveals perceptual interference between somatosensory submodalities”

We are grateful to you and to the reviewers for their thoughtful and attentive reading of the MS. Each point raised has allowed us to make a substantial improvement or useful clarification. Appended to this letter is our point-by-point response to the comments raised by the associate editor and the reviewers. We have made numerous improvements throughout to incorporate these suggestions as thoroughly as possible. The comments from the reviewers include concerns over whether sanshool-tingling might be mediated by other afferent pathways. These are important elements that are required if our central conclusion that “touch inhibits touch” is to stand. More specifically, Experiments 5 & 6 are control experiments addressing the possible afferent fibres, other than RA afferents, that might contribute to sanshool tingling. We agree with the reviewers that these controls are crucial for our overall message, and we have done our best to provide a thorough reply to reviewers’ concerns. Unfortunately, doing so took the paper over the PRSB page limit. We have therefore put Experiments 5 & 6 into Supplementary Material, leaving the main text to focus on the touch inhibits touch experiments, and just summarising the results of the control results. We can revise this if necessary and put the pathway-affinity controls back into the main text if reviewers or editor prefer. This reply letter deals fully with all the reviewers’ concerns with Control experiments 5,6.

Additionally, we have clarified our interpretation, modified the Discussion, and provided a stronger theoretical interpretation of our results. We omitted a lengthy paragraph on possible top-down perceptual filtering in sanshool tingle, since this did not seem pertinent to the main argument, and we needed to save space.

We believe that the manuscript has now been significantly improved and hope you will consider it for publication on Proceedings of the Royal Society B.

We would like to take this opportunity to express our sincere thanks to the reviewers who identified areas of our manuscript that needed corrections or modification.

Your Sincerely,

Prof. Patrick Haggard and co-authors

Institute of Cognitive Neuroscience, University College London
Address: Alexandra House, 17 Queen Square, London, WC1N 3AZ
Email: p.haggard@ucl.ac.uk
Tel: +44 (0) 207 679 1153

Reply to Associate Editor Board Member

Replies are reported immediately after each comment to facilitate reading.

Associate Editor Comments: Your manuscript has been reviewed by three experts. Although there is considerable enthusiasm for your study, several concerns have been raised. The most serious is that you do not present electrophysiological data showing that sanshool activates RA1 afferents and does so selectively. This is a significant problem. I am recommending rejection, but I would encourage you to resubmit a revised manuscript that addresses this issue.

Reply to Associate Editor: We thank the Associate Editor for providing us with an opportunity to revise our manuscript. We agree with the Editor (and Reviewer #3) that electrophysiological data would allow us to show RA1 fibre selectivity. However, this is absolutely not trivial to do. Microneurography in principle allows individual fibres to be isolated, characterised as RA/SA etc., and then recorded while experimental stimuli are applied. However, there are two barriers. First, isolating single afferents is difficult, and effectively involves blind sampling. Second, there are few research groups recording from trigeminal afferents: we have identified just 10 papers, and none in the last decade. Moreover, most of the papers we did find come from a single lab. We entirely agree with the research ambition of identifying whether sanshool activates RA1 afferents (see Discussion, ll. 349-359, ll. 371-376 and ll. 443-451), but it seems a high feasibility bar to request data that requires such a very highly specialised method. We also note that Reviewer #1's neurophysiological concerns relate more to whether our steady pressure stimulation could activate RAs, rather than whether sanshool activates them. Reviewer #3, incidentally, states "if there's no direct evidence why go there? The story is interesting even without it", meaning that they don't see the link to RA activation as essential, and consider psychophysical identification of the perceptual channel activated by sanshool as sufficient.

Furthermore, we believe that lack of electrophysiological data does not necessarily devalue our finding for the following reasons. First, we wish to distinguish between perceptual channels and the physiological afferent fibre activation. In the somatosensory system (or in any sensory system), perceptually relevant feature channels have been identified by using various psychophysical (behavioural) techniques, such as by measuring the threshold differences across different stimulus types, using methods of adaptation and masking (Gescheider et al., 2009). We believe, along with most perceptual scientists, that feature-specific perceptual channels must ultimately correspond to specific receptor types and specific afferent fibres. Sometimes this correspondence can be proven directly: for example, microstimulation of an RA1 afferent produces a vibration sensation while stimulation of an SA1 afferent produces a sensation of pressure. However, not every perceptual study provides this direct percept-afferent correspondence. Many important studies in visual (e.g. Stevens, 1957) and somatosensory (e.g. Weber, 1905) perception are based on psychophysical methods alone. Physiological and psychophysical techniques are deeply complementary, and there are several examples of physiological knowledge advance gained through psychophysics, as well as vice versa (Read, 2015).

In our study, channels are defined in the perceptual space, not in the peripheral fibre activation space. We do agree that the physiological impact of sanshool is not limited to RA fibres alone: this was already clearly stated in the manuscript (ll. 379-380). However, the *tingling percept* induced by sanshool clearly includes a component of flutter-vibration perception, which has long been associated with RA activation, as confirmed by many studies that we cite (e.g. Bryant & Mezine, 1999; Hagura et al., 2013; Kuroki et al., 2016). Furthermore, previous psychophysical studies using sanshool confirmed that sanshool tingling corresponds closely to the frequency range that defines the RA system. These studies were based on established methods of perceptual adaptation (Hagura et al., 2013) and confirmed by perceptual masking (Kuroki et al., 2016). Therefore, we may confidently say that sanshool does activate a flutter-range selective channel (i.e. the *putative* RA channel). Given this fact, our aim in this study was to investigate whether this perceptual channel is modulated by concurrent stimulation of another perceptual channel, namely the

putative SA channel. Since we know from previous masking studies that sanshool does not directly activate the steady-pressure perceptual channel associated with SA afferents (Kuroki et al., 2016), we can use psychophysical methods to test for this interaction.

We have now made this point clearer both in the Introduction (ll. 62-66; ll. 82-87) and in the Discussion (ll. 349-359; ll. 371-376; ll. 443-451). When we mention “RA channel”, we mean “putative RA channel”. We thus identify the channels perceptually, and make putative associations to specific afferent fibres, as is common in this field (for a review see Johnson & Hsiao, 1992).

Reply to Reviewer #1

Replies are reported immediately after each comment to facilitate reading.

Reviewer #1 Comments: In the present study, Cataldo and colleagues characterized the effect of sustained pressure, temperature, and anaesthesia on the tingling intensity evoked by the activation of the rapidly adapting mechanoreceptors (RA) using hydroxy-alpha-sanshool (sanshool). This experimental approach seems suitable to test the interactions across somatosensory sub-modalities, but there are important caveats that limit the conclusions. In particular, the authors claim to manipulate selectively the slowly adapting mechanoreceptors (SA) system using mechanical stimulation. However, there is no physiological evidence in the current study to confirm that the constant pressure stimuli used only activate the SA system. Because of this limitation, any conclusion the authors draw on the perception of the sanshool stimulus needs to be based on how the RA channel responded to the pairing of the mechanical stimulation and sanshool – the authors cannot definitively tease apart the influences of the mechanical test stimuli on the RA and SA systems using only their psychophysical methods. Unfortunately, the novelty of the authors' study rests on the claim of across-channel interactions, which cannot be conclusively demonstrated without neurophysiological evidence. Accordingly, we cannot recommend publication. We provide more detailed comments below that we hope can be helpful for improving future versions of the manuscript.

Reply to Reviewer #1: We thank the Reviewer for their constructive comments. In our reply to the Associate Editor above, we discuss generally the idea of perceptual channels, and their relation to physiological channels defined by different fibre classes.

The Reviewer is right that we focus on cross-channel interactions. We assume, but do not directly demonstrate, that specific perceptual channels correspond to particular physiological channels. This approach is widespread in tactile research, and is based on a large body of evidence recording from peripheral neurons (with microneurography) and central neurons.

*The Reviewer's specific concern is whether our mechanical stimuli might cause firing in RA afferents. Presumably, the reviewer believes that some RA activation might be caused by steady pressure stimuli, and this additional activation would then *mask* sanshool-evoked activation of the same RA channel, leading to attenuation of the tingle percept.*

However, RA neurons typically fire during the dynamic phases of indentation (onset/offset stimulation) only, and lack a sustained response (Johansson & Vallbo, 1983; Knibestöl, 1973). Indeed, when Knibestöl (1973) recorded RA discharge in response to dynamic and static skin indentations, he found no firing whatsoever during a static pressure phase where the probe's indentation was held constant. In contrast, SA mechanoreceptors show sustained discharge during constant skin indentation for up to 30 minutes or longer (Iggo & Muir, 1969; Johansson & Vallbo, 1983; Knibestöl, 1975). Intra-channel masking theories would then predict rapid rebound of tingle sensation as the masking effect of the hypothesised pressure-evoked RA firing decreases. In fact, we found that the attenuation of sensation remains as long as steady pressure is applied, up to 10 s, with only very modest recovery over time (Experiments 1, 2, and 4). This persistence cannot readily be explained by an intra-channel RA-only account, as the RA response to steady pressure would have effectively decayed after the transient part of the indentation.

In addition, Experiments 3 and 4 show that pressure-induced attenuation of sanshool-evoked tingling increases linearly with the indentation. Linear increase in firing rate with indentation is a characteristic marker of SA fibres (Iggo & Muir, 1969; Knibestöl, 1975), but is absent in RA fibres, which are instead mostly affected by indentation velocity (Knibestöl, 1973).

A final concern about possible RA activation from steady pressure might potentially come from the way the mechanical stimuli are delivered. We agree that our manual stimulation (Experiments 1 and 2) might be imperfect: any tremor in the experimenter's hand, for example, could result in unintended RA activation. We deal with this possibility in our reply to Reviewer #3.

We have now clarified all these important points in the Discussion (ll. 371-376; ll. 409-422; ll. 443-451; ll. 465-472).

Reviewer #1 Comments: The results from experiments 1-4 demonstrate that the perceived tingling intensity is affected by sanshool concentration, differences in mechanical pressure, and when intensity judgements are performed (relative to the mechanical pressure onset). However, there was no systematic evaluation of the dose-time-effect on the tingling perception. The best approach seems to be the experiment 4 design (tingling intensity match) but, why did the authors not characterize the frequency and amplitude that best match the tingling sensation as a function of sanshool dose/time? This step is crucial to describe the psychophysical attributes of the perception evoked by sanshool. It will help to characterize which attributes (frequency or amplitude) are affected when other stimuli are presented. And even more, it will shed light on the perceptual variability across subjects. For example, which is the best frequency and amplitude to describe the 20% sanshool effect? How reliable/variable is the effect as a function of time (i.e. there is a plateau or a peak intensity effect)? Does the frequency or the amplitude (or both) change as a function of dose and time? Is 50 Hz the best frequency to match the tingling intensity across subjects? The absence of calibration limits the interpretation of the authors' results.

*Reply to Reviewer #1: We are grateful to the Reviewer for these interesting and important questions. It is unclear whether the Reviewer is asking about the sanshool-evoked tingling, or about the attenuation thereof by steady pressure. Many of the questions about sanshool-evoked tingling have been studied in previous research. Here we are primarily interested in pressure-induced attenuation. For example, the characteristic temporal frequency of sanshool-induced tingling has been already studied (Hagura et al., 2013). Using rigorous psychophysical methods, those authors found that the perceived frequency of sanshool tingling was consistently around 50 Hz, corresponding to the range of tactile RA1 afferent fibres. Because the interaction of interest in this paper is an attenuation or inhibition, our focus is restricted to perceived *intensity* of sanshool-evoked tingle, rather than perceived frequency. This is now clearly reported in the manuscript (ll. 452-458).*

*Concerning the effect of sanshool dose on sensory amplitude, our Experiment 3 clearly shows that dosage *does* indeed affect the perceived intensity of tingling. This data was presented in Figure S2 (now Figure S1). In particular, the 80% sanshool solution produced a two-fold increase in participants' rating of tingling intensity, compared to the 20% solution (ratings for 80% solution: 6.6 on a scale from 0 to 10; ratings for 20% solution: 3.2 on a scale from 0 to 10; see Supplementary Figure S1). This is in line with previous studies showing that repeated application of a sanshool derivative evokes a progressively growing intensity of tingle sensation (Albin & Simons, 2010). However, dose-dependency of the tingling sensation was not the core goal of this experiment, so we did not systematically explore multiple dose levels. This could be done in further studies with the appropriate experimental design, as we now explicitly mention in the limitations section of the Discussion (ll. 456-458).*

The Reviewer also mentions variability across participants. Anecdotally, we have found sanshool-evoked tingling to be present in the vast majority of individuals we have tested, both in formal experiments and in informal demonstrations at science festivals. For example, over the 117 participants recruited in the present study only two (1.7%) did not report any tingling sensation (and were accordingly excluded, as reported in the Supplementary Methods). The pressure-induced attenuation of tingling was also highly consistent: of the 42 participants tested in our main Experiments 1-4, three (7.1%) did not show a reduction of tingling under static pressure stimulation.

Finally, temporal effects of sanshool have been described in both animal (Bryant & Mezine, 1999) and human (Albin & Simons, 2010) studies. Anecdotally, we have found the temporal profile of sanshool-evoked tingling quite consistent in our study: the Zanthelene sanshool solution we use takes about 1-2 minutes after application for the tingling sensation to begin, and then ramps up in intensity until reaching a plateau after a few more minutes. It then

remains fairly steady, and is still detectable after around 30 minutes. We agree that this would be an interesting area for future study, but it was not part of our scientific aim here.

The Reviewer refers to “absence of calibration”. We are not sure what this means. The Reviewer appears to be requesting a general psychophysical characterisation of sanshool-evoked sensations. The core features of the sensation have been rigorously studied before, in papers that we clearly referenced in the manuscript, by multiple research groups including our own. There appears to be good consensus on the sensory qualities of the experience. We argue that these previous studies already provide sufficient information about the sensation to allow us to focus on the specific scientific objectives of the present study. The present study does not focus on what the sanshool sensation *is*, but on how the sensation is modulated by touch.

Reviewer #1 Comments: The results from experiment 4 showed that the most prominent effect of tactile pressure on tingling intensity occurred immediately after the stimulus started. This period also happens to be the interval during which the RA and SA channels are likely to be co-activated by the mechanical stimulation. Therefore, in the absence of neurophysiological evidence, it is important to be cautious about claiming that only one group of mechanoreceptors is responsible for the perceptual effect. For example, from experiment 1 & 2, how much of the results were biased by the early part of the tactile stimulation?

Reply to Reviewer #1: This is a good point. Microneurographic studies show that RA afferents only fire during the dynamic phase of a skin indentation stimulus (Johansson & Vallbo, 1983; Knibestöl, 1973). Thus, a measure of immediate tingling attenuation could, in principle, be mediated by RAs. However, the method of adjustment used in Experiment 4 meant that participants' adjustments of mechanical vibration amplitude took several seconds (on average $2.30 \text{ s} \pm 0.62 \text{ SD}$), as the new supplementary sample video shows (Supplementary Video S1). Thus, any RA firing caused by steady pressure is likely to have ended by the time the perceptual equivalence is obtained, and the psychophysical report is recorded.

Perhaps more importantly, any RA firing due to application of steady pressure should decrease rapidly. While the Reviewer is right that the 0 s interval showed the strongest attenuation of tingle, the difference in tingle intensity between 0 s and 10 s of static touch (equivalent to $0.84 \mu\text{m}$ of vibration amplitude) was almost an order of magnitude *smaller* than the difference between the 10 s interval and the baseline tingling ($6.35 \mu\text{m}$). This means that any putative contribution of transient RA coactivation described by the Reviewer could only have a minimal effect on the final percept, compared to the contribution of the static pressure stimulus attributable to SA afferents. Thus, the fact that tingle remains strongly attenuated after 10s steady pressure, speaks strongly towards a predominantly-SA rather than a predominantly-RA mechanism of attenuation. Incidentally, a 10 s delay between pressure being applied and psychophysical judgement was also used in Experiments 1 and 2, and again showed strong attenuation of tingle. We have clarified this point in the Discussion (ll. 409-422).

We have now revised and re-plotted Figure 3 (now Figure 2E) from Experiment 4, to show the perceptually-equivalent amplitude at baseline before steady pressure is applied, as well as after (see new Figure 2E below and on p. 16 of the manuscript).

Figure 2E. Results from Experiment 4. Participants ($n = 14$) adjusted the amplitude of a mechanical vibration to the upper lip so as to match perceptual intensity of sanshool tingling on the lower lip. When varying levels of steady pressure (0.05, 0.20, 0.35 N) were additionally applied to the lower lip, the perceived amplitude of its sanshool-evoked tingle dropped rapidly, with modest but significant recovery over 10 s following pressure application. Error bars indicate the SEM across participants and coloured dots indicate individual data.

Finally, the reviewer asks whether participants judgements in Experiments 1 and 2 could be based on the early part of the trial, when attenuation is maximum following tactile onset. Participants were explicitly told to report the tingling intensity at the time when they were prompted to do so, which was 10 s from the pressure onset, as stated in the methods. Although it is theoretically possible that they were influenced by their experiences prior to the moment of being prompted, this seems unlikely since short-term memory for somatosensory stimuli is apparently limited to under 1 second (Harris et al., 2002).

Reviewer #1 Comments: Related to figure 3, why did the authors not include the amplitude before the tactile pressure? Panel B and C must show the 10 conditions (3 pressure X 3 times from onset and 1 amplitude before tactile stimulation). Also, the authors must present the data from individual subjects in all of the experiments rather than simply providing the group-averaged summary statistics. This standard practice is important for the reader to appreciate the across-subject variability/consistency. This is particularly important given the relatively low number of stimulus repetitions for particular conditions tested in some experiments.

Reply to Reviewer #1: We thank the Reviewer for this suggestion. We have now prepared a new figure, including the plot of the individual data, and pre-pressure data. Please see new Figure 2E above and on p. 16 of the manuscript.

Reviewer #1 Comments: With exception of experiments 1 and 2, either the mechanical probe or the high/low dose location was fixed across trials and subjects. Is there any reason that the test tactile/temperature/anaesthesia stimuli were always applied to the lower lip? With such a regular and consistent stimulus, it is not obvious that the subject even needs to attend to the other stimulus in the two-interval design. It would have been better to randomize the delivery of the mechanical stimulus. Furthermore, if the dose location cannot be randomized on a trial by trial basis, at least should be done across subjects.

Reply to Reviewer #1: Our choices were pragmatic. The lower lip was much more accessible to our stimulation methods than the upper lip. That our results could generalize across lips was already demonstrated in Experiments 1 and 2. Therefore, in Experiment 3, we focussed on optimizing stimulation, rather than generalization.

*We think the Reviewer has misunderstood our trial design. Experiment 3 did not use a two-interval paradigm, but a single stimulus comparison paradigm. The upper lip is continuously tingling, and acts as a reference intensity level. Participants compare the tingle felt on the lower lip to this reference. The Reviewer suggests that participants might effectively ignore the *instantaneous* value of the reference on each trial. While this is possible, it does not undermine the fact that people did experience some sanshool-evoked tingle on the upper lip in Experiment 3, just as they did in Experiment 1. While participants might have misperceived or misrepresented the instantaneous value of the reference, it is difficult to see how this could produce systematic biases in the way that steady pressure modulated the perception of the comparison stimulus. Thus, the objectives of the experiment could be achieved despite the potential difficulties the Reviewer raises. While an ideal experimental design might randomise the sanshool application from trial to trial, the slow onset and offset of the sanshool sensation makes this impractical (see reply to earlier comment).*

Reviewer #1 Comments: Also, it is important to explain why different stimulation durations were used across experiments (exp 1,2 & 4 = 10 s, exp 3 = 1s, exp 6 = 4 s)?

Reply to Reviewer #1: The differential duration of static touch across experiments was constrained by two factors: the average duration of sanshool tingling after application (~40 minutes) and the different demands of each task. For example, Experiments 1, 2, and 4, were designed to obtain multiple ratings per each trial. This allowed us to decrease the number of trials and increase the duration of the static touch stimulus. In particular, Experiment 2 comprised only 24 trials in total, allowing a relatively long (i.e. 10 s) duration of static touch stimulation. Conversely, the complex design of Experiment 3 meant that participants were required to complete a total of 150 trials in the same time window. We therefore had to substantially decrease the stimulus duration (i.e. 1 s). The choice of 4 s for Control Experiment 6 (which we have now had to move to Supplementary Material because of space constraints – see remarks to Editor) was also practical. This experiment was conducted in the context of a public science festival event that limited participant engagement to about 10 minutes. Thus, to obtain enough statistical power to compare the different experimental conditions, we opted for a combination of a few trials (12), medium stimulation duration (4 s), and a large sample size ($n = 51$).

Importantly, regardless of the reasons underlying our choices, the use of differential timings does not seem to constitute a threat to our findings, as in our opinion they rather show the generality of the effect we describe. Mechanical stimulation rapidly and enduringly attenuated sanshool-evoked tingling. We have added a short sentence in the Discussion (II. 459-464) to clarify this point.

Reviewer #1 Comments: What is the density of C-tactile fibers on the lips? The authors cite evidence that these fibers are found in the hairy skin (page 6; “In particular, C-tactile fibers are only found in hairy, but not glabrous skin”), but appear to base their conclusions on the effects of temperature manipulations on an assumption that C-fibers are also found on the lips. The assumptions of experiment 6 must be re-evaluated and clarified. Also, it is important to report the pressure applied in this experiment.

*Reply to Reviewer #1: We thank the Reviewer for pointing this out. We agree that the rationale of Control Experiment 6 was not entirely clear. We have now explained the logic and purpose of this experiment much more clearly. We have had to move the experiment to Supplementary Material because of space constraints – please see remarks to Editor. Control Experiment 6 was designed as a control or manipulation check, to *rule out* any potential contribution of C-tactile fibers to our effect. To do this, we investigated whether our basic effect showed the thermal modulation pattern that would be expected were it mediated by C-tactile fibres – but we failed to find this pattern. Results from Control Experiment 6 thus appear to rule out the possibility that unknown C-tactile fibers may mediate this effect. The Reviewer asks about the distribution of C-tactile fibres in the skin. Although C-tactile fibers are commonly found only on hairy skin, there is some electrophysiological (Djouhri, 2016) as well as psychophysical (Nagi et al., 2015; Nagi & Mahns, 2013) evidence of the existence of C low-threshold mechanoreceptors in the *glabrous* skin. C-fibres were originally identified on the face (Nordin, 1990). We are unaware of any report of C-tactile afferents from the lips. However, sampling in microneurographic studies remains a difficult issue, and we cannot exclude that C-tactile afferents may be present on the lips, nor can we exclude that such afferents are activated by our steady pressure stimuli – even though these are different from the stroking stimuli that preferentially activate C-tactile fibres in other skin regions. Nevertheless, we submit that our observed thermal modulation pattern makes explanations of sanshool-tingle based on C-tactile afferents less likely.*

Control Experiment 6 was conducted in the context of a science fair. Our computer controlled robotic arm and strain gauge setup to control pressure could not feasibly be used to control pressure. Instead, we asked participants themselves to approach the probe of the Peltier device in each trial and apply gentle touch with their lower lip for 4s. Extra care was taken in making sure that the attendees kept their lip in contact with the stimulator for the entire duration of the stimulation (4 s). Moreover, the experimenter visually monitored that a stable pressure level was maintained after the initial contact. We have clarified this point in the Methods section of the Experiment 6 (now in the Supplementary Material).

Reply to Reviewer #2

Replies are reported immediately after each comment to facilitate reading.

Reviewer #2 Comments: This paper is a joy to read. Original; informative; clearly written; with a logic sequence of experiments covering the relevant question in an adequate and ingenious way. “Chapeau...”

Reply to Reviewer #2: We thank the Reviewer for this nice comment!

Reply to Reviewer #3

Replies are reported immediately after each comment to facilitate reading.

Reviewer #3 Comments: This is an interesting and important psychophysical study showing the inhibition of chemically evoked tingling by pressure. To my knowledge, there are no direct nerve recordings (microneurography) in humans with sanshool. If the authors have electrophysiological data in humans showing that sanshool activates RA1 afferents, and does so selectively, that will really strengthen this paper. The finding that “touch inhibits touch” is interesting in its own right, but in the absence of electrophysiological data, I am not convinced with the link drawn between stimulus/percept and the afferent type. Let’s take the RAs first. If this presumed link is based on intraneural microstimulation studies, these have shown that the percept evoked by activation of RA afferents is frequency-dependent, and at higher stimulation frequencies, the percept evoked by activation of RA1 or RA2/PC afferent is qualitatively indiscernible (Ochoa and Torebjörk 1983, J Physiol). Most microstimulation studies were performed on the glabrous skin of the hand, so whether those observations can be extrapolated to other body regions, like the lip, in this case, is unclear. Also, the idea that the RA1s alone are responsible for encoding flutter-range frequency has been challenged in a recent paper showing that the PCs can do that just as well (Birznieks et al. 2019, eLife), indicating that it is the spiking pattern, and not the receptor type, that determines frequency perception. The same argument applies to sustained indentation. The authors refer to sustained indentation as SA1-mediated, but SA1s are not the only afferent type that responds to sustained indentation. SA2s obviously have a sustained response, and at higher indentation forces, even field afferents show some sustained activity (Nagi et al. 2019, Sci Adv). Also, CTs have some sustained response.

*Reply to Reviewer #3: We thank the Reviewer for their comments. As we have replied to the associate editor and Reviewer 1, this paper focuses on *perceptual* channels, which can be defined psychophysically. Our target is thus the channel that induces flutter-range percept. We do not claim that sanshool activates only one specific physiological afferent type, nor that this one specific afferent is suppressed by pressure input. Our claim was to say that, flutter-range vibration channel interacts with the pressure input. We regret that in our previous manuscript, the distinction between information-based and physiological receptor-based or afferent-based definitions of channels was rather unclear. Now, we have made clear that, when we mention “RA channel”, it is about “putative RA channel” identified through psychophysical methods.*

Reviewer #3 Comments: Other comments/suggestions:

Vibration thresholds decrease as the pressure of applied stimulus is increased (e.g. Lowenthal et al. 1987, Diabetes Care). This should be discussed in the context of the findings of the current study.

Reply to Reviewer #3: We thank the Reviewer for flagging this up. We have now added a new paragraph in our manuscript to discuss our results in view of these relevant previous findings (ll. 487-498).

Reviewer #3 Comments: Weerakkody et al. (2007, J Physiol) showed an impairment of proprioceptive acuity when high-frequency vibration was applied but not when low-frequency vibration was applied. Proprioception has a cutaneous component, some even consider it to be part of the tactile domain, and these interactions I think are relevant and should be acknowledged here.

Reply to Reviewer #3: We do not entirely follow the Reviewer's point. We agree that proprioception has a cutaneous component, but we do not see how this is relevant in the context of our study. The study cited could explain that different afferent classes contribute differentially to proprioception – but does not seem to directly clarify whether different mechanoreceptor channels interact or not. Our study did not aim to investigate proprioception, nor to provide an exhaustive account of any bodily sensation.

Reviewer #3 Comments: “Experiment 1 the lips were manually stimulated by the experimenter with a probe (diameter: 14 mm, force: ~1.5 N) for 10 seconds.” This is handheld without force-feedback control – RAs, for instance, are exquisitely sensitive to dynamic changes, and any small movement that may or may not be perceptible could make the RAs fire.

Reply to Reviewer #3: We agree with the Reviewer that – despite being a very common practice (e.g. see von Frey filaments, graphesthesia stimuli, two-points discrimination test, etc) – hand-delivered mechanical stimuli do not provide a good control over the pressure level exerted. We think this is similar to the concern raised by Reviewer #1 regarding possible RA activation by our pressure stimuli. For this reason, Experiments 3 and 4 used precise robot-controlled mechanical stimuli, and additionally stabilised the participant themselves using a chinrest. Both these design features should minimise micromovements. We still found a strong inhibition of tingling due to static pressure, clearly replicating the results from Experiments 1 and 2. We have added a sentence describing this possibility (ll. 465-472).

Reviewer #3 Comments: Please provide the corresponding value in pressure for all mechanical stimulations.

Reply to Reviewer #3: Thanks for this point. We report contact force where this is known. In Experiments 1 and 2, pressure stimuli were delivered through a custom-made spring-loaded stimulator, which was previously calibrated to apply a force of ~1.5 N (see Methods, ll. 113-114). In Experiments 3 and 4, contact force was precisely controlled, and is reported (ll. 156-157 and ll. 181-182). Experiment 5 (now moved to the Supplementary Material because of space constraints – please see remarks to Editor) is a control experiment without contact force stimulation. Experiment 6 (now moved to the Supplementary Material because of space constraints – please see remarks to Editor) focussed on thermal modulation rather than on force control. The testing environment (at a Science Fair) made the force-control setup of experiments 3-4 unfeasible. Instead we asked participants themselves to approach the probe of the Peltier device in each trial and apply gentle touch with the lower lip for 4s. Extra care was taken in making sure that the attendees kept their lip in contact with the stimulator for the entire duration of the stimulation (4 s). Moreover, the experimenter visually controlled that after contact a stable pressure level was kept throughout. We have clarified this point in the Methods section of the Experiment 6 (now in the Supplementary Material).

Reviewer #3 Comments: “Participants rated the tingling sensation relative to its intensity during the no-pressure baseline: 0 as no sensation, 10 as same level as the intensity during baseline, and rating above 10 as higher intensity than the baseline period.” Can you please clarify this?

Reply to Reviewer #3: We apologise for not being clear. In the baseline condition before the application of the pressure probe, participants experienced the sanshool tingling sensation. They were asked to memorise this baseline intensity. After that, the probe was applied, and the participants again experienced the tingling sensation. Then, they were asked to rate the experienced tingling sensation while the probe was applied relative to the baseline. They were asked to use the rating scale as follows: a rating of 10 would indicate the same level as the intensity during baseline, 0 would indicate no tingling sensation, and ratings above 10 would indicate intensities higher than during the baseline period. We have clarified this in the Methods section (ll. 119-124).

Reviewer #3 Comments: “The rating occurred after 10 s of continuous probe touch to avoid any mechanical transient effect.” Please refer to my comment above about micro-movements. It is handheld stimulation so almost impossible to avoid these.

Reply to Reviewer #3: We agree with the Reviewer about this point, and explicitly mention this possibility in the discussion section (ll. 465-472). However, as we note, better force control in Experiment 3 did not destroy the effect. See also our previous reply.

Reviewer #3 Comments: “Each trial started when the participant confirmed that the tingling sensation had returned, typically after a few seconds.” This is quite interesting that the inhibition outlasts the stimulus by a few seconds. What does this depend on?

Reply to Reviewer #3: We share the same curiosity as the Reviewer on this point. One speculative explanation may come from microneurographic observations that many mechanoreceptive afferents transiently increase firing at the offset of a skin indentation. For example, Furusawa and colleagues (1992) described a type of mechanoreceptive afferent in the human lips and oral mucosa that showed a sustained off response starting at the offset of the stimulation and lasting approximately 500 msec. Clearly, this issue should be directly addressed by future studies. The psychophysical methods that we used here may lack the temporal resolution required to track the dynamics of the returning tingling sensation, so we hesitate to make any strong assertion on the basis of the current data.

Reviewer #3 Comments: I would suggest including a methods figure. There are six experiments, and I think a schematic representation of these would be helpful.

Reply to Reviewer #3: We thank the Reviewer for this useful suggestion. We have now added a new method figure with the details of the different experiments (see new Figure 1 on p. 10 and below).

Figure 1. Experimental methods. A-B: In Experiments 1 and 2, participants ($n = 10$ in each experiment) experienced sanshool tingling all over the lips, while sustained touch (~ 1.5 N) was manually applied in different locations for 10 s. Participants reported the effect of touch location on sanshool tingling by rating the change in tingling intensity on the centre of the lower lip (A: Experiment 1) or all over the lips (B: Experiment 2). C: In Experiments 3, weaker and stronger sanshool solutions caused weaker and stronger tingling intensities on the upper and lower lips, respectively. Different levels of sustained force (0.05, 0.1625, 0.275, 0.375, and 0.5 N) were then applied to the lower lip by a closed-loop robotic device (Supplementary Figure S2). Participants ($n = 8$) reported which lip had the strongest tingling, as a function of sustained force. D: In Experiment 4, participants ($n = 14$) estimated the intensity of sanshool tingling on the lower lip by adjusting the amplitude of 50 Hz vibration applied to the upper lip until the intensities felt equal. Meanwhile, different levels of sustained force (0.05, 0.20 or 0.35 N) were applied on the lower lip. The adjustment was done at four different timings from the onset of the force (before pressure, and at 0, 5, and 10 s after pressure) (see Supplementary Video S1 for an example of a trial).

Reviewer #3 Comments: “The experiment consisted of six blocks where static touch was applied twice to central positions 3 and 6 and once to the other areas (ten ratings from each participant per session).” Why the difference? What was the rationale for that?

Reply to Reviewer #3: We anticipated that the local effects of touch on tingling would exceed the remote effects. To have increased sensitivity for the more important local effects, and the homologous location on the upper lip, we oversampled there. We have now mentioned this in the manuscript (ll. 127-130).

Reviewer #3 Comments: Referral to the figures does not follow the sequence in the text, for example, Fig 1 A, then 1 C...

Reply to Reviewer #3: Thanks for noticing, and we apologise for this error. This has now been corrected.

Reviewer #3 Comments: “Yet, instead of only rating the sanshool tingling at a fixed location, participants rated the tingling intensity for all four lip quadrants.” Can you explain this more? Why were the instructions changed in Experiment 2?

Reply to Reviewer #3: *In Experiment 1, participants judged the tingling on only one fixed location. This procedure requires participants to sustain their attention to one fixed location on the lips throughout the task. To make sure that the effect obtained in Experiment 1 is not due to such sustained spatial attention, in Experiment 2, participants judged the tingling of all four locations on the lips, of which one was touched by the probe. We have clarified this point in the Methods (ll. 136-141).*

Reviewer #3 Comments: “Forces were applied by a closed-loop system composed by a linear actuator (ZABER, XYZ Series, Vancouver, Canada) and a force gauge (McMesin, PFI-200N GEB, Slinfold, UK) (Figure 2), which continuously maintained the desired pressure level.” Please provide an example (force trace).

Reply to Reviewer #3: *Thanks, we have now added a new supplementary figure (see below and Supplementary Figure S3) with the force traces from three trials of a representative participant in Experiment 4.*

Supplementary Figure S3. *Force traces from three trials of a representative participant in Experiment 4. The continuous coloured lines represent the reading of the Mecmesin force sensor from the moment of contact between the probe and the lips until the end of the trial. Three trials are shown, with force target levels of 0.35 N (purple trace), 0.2 N (red trace), and 0.05 N (pink trace). The dashed portions of each trace represent interpolated data for the period in which participants adjusted a mechanical vibration to the upper lip to match the intensity of sanshool-evoked tingle on the lower lip. During this adjustment period the force could not be recorded. The duration of each adjustment depended on the participant themselves (see Methods and Supplementary Video S1). Thin dashed horizontal lines indicate the target force for each trial. Surrounding coloured bands indicate the resolution of the Mecmesin PFI-200N (± 0.04 N) according to the manufacturer’s technical specification.*

Reviewer #3 Comments: “To rule out the contribution of nociceptive C-fibres to sanshool tingling, we measured both pain thresholds and perceived intensity of tingling before and after blocking the activity of small fibres through lidocaine.” How was the contribution of A-beta nociceptors (Nagi et al. 2019, Sci Adv) ruled out?

Reply to Reviewer #3: Thank you for this useful point. The focus of Experiment 6 (now moved to Supplementary Material because of space constraints – please see remarks to Editor) is to exclude the contribution of a nociceptive channel to sanshool tingling. But as the Reviewer points out, A β fibres can also contribute to the nociceptive system, in addition to C-fibres. Further, A β fibres might also be affected by lidocaine. In general, however, lidocaine is thought to preferentially affect small fibres, particularly initially (Paqueron et al., 2003). Our data were collected during this time-window. However, we cannot entirely exclude an A β component, and we have accordingly mentioned this possibility in the text (ll. 380-381).

Reviewer #3 Comments: “In particular, C-tactile fibres are only found in hairy, but not glabrous skin”. I would suggest revising this to say that there is no evidence of a phenotypically identical class of hairy-skin C-tactile fibers in human glabrous skin. Just to clarify my point: there is electrophysiological evidence of the existence of C low-threshold mechanoreceptors in the glabrous skin of rat hind paw (Djoughri 2016, Neurosci Lett), and psychophysical evidence, using nerve blocks, of a C low-threshold input from the glabrous skin of human hand (Nagi and Mahns, 2013, Exp Br Res; Nagi et al. 2015, BMC Neurosci) .

Reply to Reviewer #3: We thank the Reviewer for this useful suggestion. We have now added a short sentence in the Methods of Control Experiment 6 (now in Supplementary Material) to clarify this point.

Reviewer #3 Comments: “Importantly, C-tactile fibres also show preference for neutral, skin temperature (32 °C) stimuli, rather than warm (40 °C) or cold (18 °C) stimuli [31]. Thus, if the sanshool tingle is mediated by C-tactile fibres, the perceived intensity of tingling should then be maximal at neutral temperatures and reduced during cold or warm stimulation (i.e. inverted U-shape).” It’s a very subtle effect of temperature on CT brush responsiveness, and the inverted U is still produced.

*Reply to Reviewer #3: We are not sure we have understood the Reviewer’s point. In previous studies, the CT fibre response as a function of temperature of the stimulating probe shows an inverted-U shape, and the effect size is large (e.g., $F(2,57.5) = 8.99$; $\eta^2 = 0.28$) (Ackerley et al., 2014). Our modulation of tingling inhibition by temperature did *not* show this inverted U shape as a function of temperature, which makes it less likely that inhibition of sanshool-evoked tingling reflects CT fibre firing. Is the reviewer perhaps confusing the inverted U shape for CT firing as a function of *temperature* with the much better-known inverted U shape for CT firing as a function of motion *velocity*?*

Reviewer #3 Comments: “Instead, the interaction appears to occur at some neural processing level where afferents from the two mechanoreceptors are integrated in a spatially organised manner.” For circuit mechanisms of presynaptic inhibition have a look at the paper by Zimmerman et al. 2019 (Neuron).

Reply to Reviewer #3: Thanks, we have now added a sentence referencing this very interesting paper (ll. 473-475).

Reviewer #3 Comments: For the bar charts, can you please include/superimpose the individual data points as well?

Reply to Reviewer #3: Thanks, now done.

Reviewer #3 Comments: “Cold touch produced the lowest ratings ($1.85 \pm \text{SD } 1.7$).” There is some evidence that CTs respond to cooling e.g. Nordin 1990 (J Physiol), Samour et al. 2015 (Pain). So could an alternative interpretation be that when CTs are co-stimulated with pressure and cold, the inhibition of tingling is most robust? Of course, the same argument applies to SAs in general which the authors make for type 1 SA, and these were indeed referred to as spurious thermoreceptors by Iggo.

Reply to Reviewer #3: If we are understanding the Reviewer's point correctly, the Reviewer is raising an alternative possibility of cold pressure probe inhibiting the tingling. However, this explanation cannot explain the enhancement of tingling during warm pressure. Further, other studies found that lower temperatures reduced CT firing (Ackerley et al., 2014), contrary to Nordin et al. (1990). We have added a sentence to explain that thermal sensitivity of CT fibres remains controversial, and we now reference the studies mentioned by the Reviewer (ll. 392-394).

Reviewer #3 Comments: “This suppression pattern clearly differs to the inverted U-shape that would be expected from C-tactile fibre thermal sensitivity.” Can you clarify this? The inverted U is not for static touch, it's actually not known.

*Reply to Reviewer #3: We think there might be a misunderstanding here. We think the Reviewer is referring to the inverted U firing profile of C-tactile fibers as a function of stimulation *velocity*. Instead, the inverted U profile we refer to is the C-tactile discharge in function of *temperature* (Ackerley et al., 2014). Ackerley and colleagues (2014) have shown that C-tactile fibers are not linearly modulated by temperature, but in an inverted U shape (i.e. maximal C-tactile fibers activation for neutral stimuli, minimal for both cold and warm stimuli). The Reviewer's reference to “static touch” seems relevant, however. Our mechanical stimuli were indeed static, and the normal optimal stimulus for C-tactile fibres is a moving stimulus. The Reviewer is right that Ackerley's stimuli were motion stimuli. Thus, the thermal modulation of CT firing could perhaps differ for static vs moving stimuli, though we do not know of any physiological reason why this should be so. Further, Ackerley et al's inverted-U function of temperature was invariant from very low velocities (0.3 cm/s) to much higher velocities. Thus, we are not sure what we can usefully make changes in the text here.*

Reviewer #3 Comments: “although we cannot directly measure RA afferent responses to sanshool, we may nevertheless use sanshool tingling sensation as a proxy for RA activation.” OK good that it's mentioned here. My point is the same – if there is no direct evidence why go there? The story is interesting even without it.

Reply to Reviewer #3: Thanks very much for this comment. As described in the answer to the comments above, throughout the revised manuscript we now focus on the perceptual feature channel, not the peripheral RA activation per se. We have modified our manuscript to show that the perceptual channel associated with the tingling sensation may be physiologically instantiated by a putative RA channel, though we cannot know this directly (ll. 62-66; ll. 82-87; ll. 349-359; ll. 371-376; ll. 443-451). We agree with the Reviewer that the psychophysical results alone are informative, even if the afferent classes associated with the sensation cannot be identified physiologically (see also comments to the Associate Editor and Reviewer #1).

Reviewer #3 Comments: In terms of the quality of the sensation, how similar is sanshool to low-frequency vibration?

Reply to Reviewer #3: Sanshool tingling has been characterised in depth by previous studies from our lab and other groups, and is consistently associated to low-frequency mechanical stimuli. For example, Bryant and Mezine (1999) reported that:

“the tingling sensation from HO α -S [i.e. sanshool] was more similar to a mild electric shock (5–7 V, either DC or AC) or a weakly carbonated solution applied to the tongue”.

In our own experience, common descriptions of sanshool tingling from naïve participants often refer to: Szechuan cuisine, pins and needles, insects crawling on the lips, or “embouchure collapse” during brass playing. Moreover, in a previous study (Hagura et al., 2013) using rigorous psychophysical methods, we have found that sanshool tingling was consistently associated with a mechanical vibration of 50 Hz (\pm 2.4 Hz S.E.).

A previous qualitative study invited participants to rate sanshool-evoked sensations according to adjectival descriptors – tingling was the dominant response (Albin & Simons, 2010). However, that study was performed on the tongue, where the sensation seems anecdotally different from the lips, and the receptor populations are different.

Reviewer #3 Comments: “Top-down filtering accounts require the implausible assumption that cold touch is more salient than warm touch to explain this result.” Why is this implausible?

Reply to Reviewer #3: Thanks. We agree this passage was not very clear. Our point is that there is no reason to assume that cold touch is more salient than the warm touch, as the hypothesis being discussed at this point would require. We have now omitted the lengthy paragraph on possible top-down perceptual filtering in sanshool tingle, since this did not seem pertinent to the main argument, and we needed to save space (see remarks to Editor).

Reviewer #3 Comments: In Experiment 2, the modulation is not spatially constrained. What are the limits on that? Can touch on the forearm inhibit tingling sensation on the lip? Like it has been shown for vibration in TMJ (Fillingim et al. 1998, Pain) although that was in the context of gain of function (allodynia). On that note actually, there’s no data for other body parts for sanshool shown here. Has it been done in an earlier study? If so, how similar is the sensation to the lip? I think this question is important for the generalizability of these results.

Reply to Reviewer #3: We thank the reviewer for raising this point. In a previous study (Kuroki et al., 2016), sanshool was applied on the fingertip, and participants experienced a tingling sensation similar to that experienced on the lips. They were able to match the sanshool-induced fingertip tingling sensation with a mechanical vibration at appropriate frequency. Therefore, the sensation induced by sanshool is generalizable across different effectors. Several studies have investigated lingual sensations to sanshool (e.g. Albin & Simons, 2010). In our anecdotal experience, these also involve a kind of tingling, but the experience on the tongue is qualitatively different from that on the lips.

We have not systematically studied how tingling on the lips is influenced by touch on a remote body part. However, our anecdotal experience has been that manipulating objects in the hand does not attenuate tingle on the lips. Further, our Experiment 2 suggests that attenuation of tingle sensation was limited to the immediate area of mechanical stimulation.

References

- Ackerley, R., Wasling, H. B., Liljencrantz, J., Olausson, H. akan, Johnson, R. D., & Wessberg, J. (2014). Human C-tactile afferents are tuned to the temperature of a skin-stroking caress. *Journal of Neuroscience*, 34(8), 2879–2883.
- Albin, K. C., & Simons, C. T. (2010). Psychophysical evaluation of a sanshool derivative (alkylamide) and the elucidation of mechanisms subserving tingle. *PLoS One*, 5(3), e9520.
- Bryant, B. P., & Mezine, I. (1999). Alkylamides that produce tingling paresthesia activate tactile and thermal trigeminal neurons. *Brain Research*, 842(2), 452–460. [https://doi.org/10.1016/S0006-8993\(99\)01878-8](https://doi.org/10.1016/S0006-8993(99)01878-8)
- Djouhri, L. (2016). Electrophysiological evidence for the existence of a rare population of C-fiber low threshold mechanoreceptive (C-LTM) neurons in glabrous skin of the rat hindpaw. *Neuroscience Letters*, 613, 25–29. <https://doi.org/10.1016/j.neulet.2015.12.040>
- Furusawa, K., Yamaoka, M., Ichikawa, N., & Kumai, T. (1992). Airflow receptors in the lip and buccal mucosa. *Brain Research Bulletin*, 29(1), 69–74. [https://doi.org/10.1016/0361-9230\(92\)90010-U](https://doi.org/10.1016/0361-9230(92)90010-U)
- Gescheider, G., Wright, J., & Verrillo, R. (2009). *Information-processing channels in the tactile sensory system: A psychophysical and physiological analysis* New York. Taylor & Francis Group, LLC.
- Hagura, N., Barber, H., & Haggard, P. (2013). Food vibrations: Asian spice sets lips trembling. *Proceedings of the Royal Society B: Biological Sciences*, 280(1770), 20131680–20131680. <https://doi.org/10.1098/rspb.2013.1680>
- Harris, J. A., Miniussi, C., Harris, I. M., & Diamond, M. E. (2002). Transient Storage of a Tactile Memory Trace in Primary Somatosensory Cortex. *Journal of Neuroscience*, 22(19), 8720–8725. <https://doi.org/10.1523/JNEUROSCI.22-19-08720.2002>
- Iggo, A., & Muir, A. R. (1969). The structure and function of a slowly adapting touch corpuscle in hairy skin. *The Journal of Physiology*, 200(3), 763–796.
- Johansson, R. S., & Vallbo, Å. B. (1983). Tactile sensory coding in the glabrous skin of the human hand. *Trends in Neurosciences*, 6, 27–32. [https://doi.org/10.1016/0166-2236\(83\)90011-5](https://doi.org/10.1016/0166-2236(83)90011-5)
- Johnson, K. O., & Hsiao, S. S. (1992). Neural mechanisms of tactual form and texture perception. *Annual Review of Neuroscience*, 15, 227–250. Scopus.

- Knibestöl, M. (1973). *Stimulus—Response functions of rapidly adapting mechanoreceptors in the human glabrous skin area. The Journal of Physiology, 232(3), 427–452.*
<https://doi.org/10.1113/jphysiol.1973.sp010279>
- Knibestöl, M. (1975). *Stimulus-response functions of slowly adapting mechanoreceptors in the human glabrous skin area. The Journal of Physiology, 245(1), 63–80.*
<https://doi.org/10.1113/jphysiol.1975.sp010835>
- Kuroki, S., Hagura, N., Nishida, S., Haggard, P., & Watanabe, J. (2016). *Sanshool on The Fingertip Interferes with Vibration Detection in a Rapidly-Adapting (RA) Tactile Channel. PLOS ONE, 11(12), e0165842.* <https://doi.org/10.1371/journal.pone.0165842>
- Nagi, S. S., Dunn, J. S., Birznieks, I., Vickery, R. M., & Mahns, D. A. (2015). *The effects of preferential A- and C-fibre blocks and T-type calcium channel antagonist on detection of low-force monofilaments in healthy human participants. BMC Neuroscience, 16(1), 52.*
<https://doi.org/10.1186/s12868-015-0190-2>
- Nagi, S. S., & Mahns, D. A. (2013). *Mechanical allodynia in human glabrous skin mediated by low-threshold cutaneous mechanoreceptors with unmyelinated fibres. Experimental Brain Research, 231(2), 139–151.* <https://doi.org/10.1007/s00221-013-3677-z>
- Nordin, M. (1990). *Low-threshold mechanoreceptive and nociceptive units with unmyelinated (C) fibres in the human supraorbital nerve. The Journal of Physiology, 426(1), 229–240.*
- Paqueron, X., Leguen, M., Rosenthal, D., Coriat, P., Willer, J. C., & Danziger, N. (2003). *The phenomenology of body image distortions induced by regional anaesthesia. Brain, 126(3), 702–712.* <https://doi.org/10.1093/brain/awg063>
- Read, J. C. A. (2015). *The place of human psychophysics in modern neuroscience. Neuroscience, 296, 116–129.* <https://doi.org/10.1016/j.neuroscience.2014.05.036>
- Stevens, S. S. (1957). *On the psychophysical law. Psychological Review, 64(3), 153–181. Scopus.*
<https://doi.org/10.1037/h0046162>
- Weber, E. H. (1905). *Tastsinn und gemeingefühl. W. Engelmann.*